

# Evaluation of UKESM aerosol size and composition using ATom measurements indicates missing marine aerosol formation mechanisms

Xu-Cheng He[1,2,3,4], Nathan Luke Abraham[1,5], Han Ding[6], Maria R. Russo[1,5], Daniel P. Grosvenor[5,7], Yao Ge[1], Xuemei Wang[3,8], Anthony C. Jones[9], Pedro Campuzano-Jost[10], Benjamin Nault[11,12], Agnieszka Kupc[13], Donald Blake[14], Jose L. Jimenez[10], Christina J. Williamson[4], Kenneth S. Carslaw[15], James Weber[16], Alexander T. Archibald[1,5], and Hamish Gordon[3]

[1]Yusuf Hamied Department of Chemistry, University of Cambridge, Cambridge, UK
[2]Institute for Atmospheric and Earth System Research (INAR), University of Helsinki, Helsinki, Finland
[3]Department of Chemical Engineering and Center for Atmospheric Particle Studies, Carnegie Mellon University, Pittsburgh, USA
[4]Finnish Meteorological Institute, Helsinki, Finland
[5]National Centre for Atmospheric Science, UK
[6]Department of Mechanical Engineering and Center for Atmospheric Particle Studies, Carnegie Mellon University, Pittsburgh, USA
[7]Centre for Environmental Modelling And Computation, University of Leeds, Leeds, UK
[8]Koninklijk Nederlands Meteorologisch Instituut, De Bilt, The Netherlands
[9]Met Office Hadley Centre, Exeter, UK
[10]Cooperative Institute for Research in Environmental Sciences (CIRES) and Department of Chemistry , University of Colorado Boulder, Boulder, USA
[11]Department of Environmental Health and Engineering, Johns Hopkins University, Baltimore, USA
[12]Center for Aerosol and Cloud Chemistry, Aerodyne Research Inc., Billerica, USA
[13]Aerosol Physics and Environmental Physics, Faculty of Physics, University of Vienna, Vienna, Austria
[14]Department of Chemistry, University of California Irvine, Irvine, USA
[15]Institute for Climate and Atmospheric Science, School of Earth and Environment, University of Leeds, Leeds, UK
[16]Department of Meteorology, University of Reading, Reading, UK

**Correspondence:** Nathan Luke Abraham (nla27@cam.ac.uk) and Alexander T. Archibald (ata27@cam.ac.uk)

**Abstract.**

Atmospheric aerosols influence climate through their interactions with radiation and clouds, yet large uncertainties remain in their simulation by global models. This study evaluates the United Kingdom Earth System Model version 1.1 (UKESM1.1) using global-scale aircraft observations from the Atmospheric Tomography (ATom) mission, focusing on aerosol lifecycle pro-

cesses in the remote marine atmosphere. We assess model performance in simulating aerosol precursor vapours, number size distributions, chemical composition, and environmental conditions. Several process improvements are tested, including sulfuric acid–ammonia nucleation, ammonium nitrate scheme, methanesulfonic acid condensation, and low-temperature isoprene-derived secondary organic aerosol formation.

Model biases differ significantly between the upper troposphere (UT) and the marine boundary layer (MBL). In the UT,

UKESM1.1 overestimates nucleation and Aitken mode particles while underestimating accumulation mode, indicating insuf-





ficient growth. In the MBL, the model overestimates primary aerosols (e.g. seasalt) and precursor gases but underestimates nucleation and Aitken mode particles, even after incorporating updated nucleation and ammonium nitrate scheme. The persistence of low aerosol number concentrations, despite overestimated precursors, suggests missing formation pathways likely involving other species such as iodine, amines, or organic vapours.

These limitations result in an unbalanced cloud condensation nuclei budget that over-replies on primary emissions. Sensitivity tests reveal that model outputs are strongly influenced by dimethyl sulfide emissions and vapour condensation schemes. Our results highlight the need for future model development to prioritise mechanistic representation of currently missing aerosol sources, rather than relying on empirical tuning, to improve aerosol–climate interaction estimates.

## 1 Introduction

Aerosols profoundly influence the Earth's climate system (Charlson et al., 1992). Depending on their chemical composition and size, they can directly affect the planetary radiation balance by scattering (e.g. sulfate aerosols) or absorbing (e.g. black carbon aerosols) solar radiation. Furthermore, aerosols act indirectly by serving as cloud condensation nuclei (CCN), thereby altering cloud microphysical properties, lifetime, and albedo, which ultimately impacts Earth's radiation balance (Twomey, 1977). While satellite observations help constrain the contemporary radiation balance, understanding the net effect of aerosols
on Earth's radiative balance since the preindustrial era, known as aerosol forcing, presents a significant challenge. This aerosol forcing quantification requires knowledge of the aerosol state in the preindustrial era, a period without direct observations.

Consequently, climate models are almost exclusively used to infer preindustrial aerosol states. These models are typically evaluated against the current aerosol state, under the assumption that present-day processes can be extrapolated backward in time. However, uncertainties associated with this assumption, coupled with limitations in accurately simulating even the
present-day aerosol lifecycle, lead to substantial uncertainties in estimates of both direct and indirect aerosol forcings (Masson-Delmotte et al., 2021).

Despite the urgent need for reducing aerosol forcing uncertainties, they have shown little improvement over recent decades. Multiple factors contribute to these persistent uncertainties. These include an incomplete understanding of aerosol emission sources and airborne aerosol formation pathways (Kirkby et al., 2023), the intricate nature of aerosol-cloud interactions (Bel-
louin et al., 2020), difficulties in representing complex aerosol processes and natural feedback loops within Earth system models (ESMs) (Thornhill et al., 2021), and, critically, the poorly constrained preindustrial aerosol state (Carslaw et al., 2013, 2017).

A prerequisite for reducing these uncertainties and ultimately quantifying aerosol-cloud-climate interactions is the accurate quantification of aerosol sources. Aerosols originate from two main pathways: direct emission of primary aerosols (e.g. dust, seasalt, primary organic aerosols and black carbon) and airborne aerosol formation (secondary aerosols) from precursor
vapours. This airborne aerosol formation pathway, involving the nucleation of low-volatility molecules followed by growth through condensation or coagulation, is estimated to contribute over 50 % of global tropospheric aerosol number concentration and dominates in the upper troposphere (UT) (Gordon et al., 2017). Therefore, a thorough understanding of airborne aerosol formation is crucial for constraining the global aerosol number budget and improving forcing estimates.



Historically, research into vapours driving aerosol nucleation has centred on sulfuric acid ($H_2SO_4$), formed via the oxidation of sulfur dioxide ($SO_2$) by the hydroxyl radical (OH) (Kulmala et al., 1998; Kirkby et al., 2011). Sulfur sources include natural emissions (e.g. from dimethyl sulfide, DMS) and anthropogenic activities ($SO_2$), making $H_2SO_4$ nucleation key to understanding both preindustrial conditions and anthropogenic aerosol forcing. Aerosol nucleation involving $H_2SO_4$ is typically assisted by water vapour and atmospheric ions, with rates highly sensitive to temperature, humidity, and ion concentrations (Kulmala et al., 1998; Vehkamäki et al., 2002; Lee et al., 2003). However, $H_2SO_4$-$H_2O$ nucleation, even considering ions, often fails to explain observed aerosol nucleation rates in the boundary layer (Kirkby et al., 2011; Dunne et al., 2016). Subsequent laboratory and field studies have established that alkaline vapours, particularly ammonia ($NH_3$) and amines (e.g. dimethyl amine, DMA), dramatically enhance $H_2SO_4$ nucleation rates (Kirkby et al., 2011; Almeida et al., 2013; Jen et al., 2014). This base-enhanced nucleation is observed across diverse environments, from polluted regions (Yan et al., 2021) to pristine polar and marine atmospheres (Jokinen et al., 2018; Beck et al., 2021; Brean et al., 2021).

Beyond $H_2SO_4$, recent research has highlighted other significant contributors to regional and potentially global new particle formation (NPF). Oxidation products of monoterpenes are major drivers in boreal forests (Yokouchi and Ambe, 1985; Ehn et al., 2014; Tunved et al., 2006; Kirkby et al., 2016), while isoprene oxidation products are implicated in nucleation events observed in the tropical UT (Andreae et al., 2018; Shen et al., 2024; Curtius et al., 2024). Furthermore, iodine oxoacids play a dominant role in coastal (Hoffmann et al., 2001; O'Dowd et al., 2002; Sipilä et al., 2016) and polar regions (Baccarini et al., 2020; Price et al., 2023). Their widespread presence and rapid nucleation kinetics suggest a broader contribution to marine aerosol formation (He et al., 2021a, b, 2023). These mechanisms can dominate regionally or act synergistically with $H_2SO_4$, collectively influencing global aerosol populations (Lehtipalo et al., 2018; He et al., 2023; Shen et al., 2024).

The formation pathways involving organic vapours and iodine species are particularly important in pristine regions. In these environments, cloud formation is highly sensitive to small changes in CCN concentrations (Carslaw et al., 2013; Koren et al., 2014). Moreover, as global anthropogenic sulfur emissions decline, the relative contribution of $H_2SO_4$-driven nucleation is expected to decrease, while the importance of organic and iodine-based mechanisms for the global aerosol budget will likely increase (He et al., 2021b, 2023). Accurately representing these processes in climate models is therefore essential for defining the preindustrial baseline, constraining anthropogenic aerosol forcing, and projecting future climate change.

Despite their climatic importance, aerosol processes remain crudely represented in many ESMs. A significant number of models used in initiatives like the Coupled Model Intercomparison Project Phase 6 (CMIP6) employ prescribed aerosol fields, which are not interactively coupled with model chemistry and climate dynamics (Wang et al., 2022). This limits their ability to simulate aerosol forcings and feedbacks realistically. Even in ESMs with interactive aerosol modules (Thornhill et al., 2021), the representation of airborne aerosol formation is often restricted to simplified $H_2SO_4 - H_2O$ nucleation schemes based on theoretical rates (Kulmala et al., 1998; Vehkamäki et al., 2002). These schemes face a fundamental paradox: while experimental and observational evidence clearly demonstrates that $H_2SO_4 - H_2O$ nucleation mechanisms alone are insufficient to explain observed aerosol number concentrations (Kirkby et al., 2011; Dunne et al., 2016; Kirkby et al., 2023), the most popular theoretical nucleation rates currently adopted in models actually overestimate aerosol nucleation by several orders of magnitude (Yu et al., 2020). This overestimation may unintentionally compensate for missing nucleation mechanisms involving other



species such as ions from cosmic rays or radon, NH$_3$, organics, and iodine compounds (Kirkby et al., 2011, 2016; He et al.,
2021b, 2023; Shen et al., 2024), but it prevents models from capturing the underlying physical processes and their associated
climate feedbacks related to nitrogen, carbon, and iodine cycles.

Robust evaluation against observations therefore is paramount for assessing ESMs performance and improving the necessary
aerosol precursor chemistry and microphysics. While satellite observations provide valuable global context, they have limita-
tions, particularly in detecting aerosols smaller than 100 nm and those near or below clouds (Seinfeld et al., 2016). Aircraft
campaigns offer complementary, high-resolution in-situ data on precursor gases, detailed aerosol number size distributions, and
sometimes chemical composition. Among available aircraft campaigns, the Atmospheric Tomography (ATom) mission stands
out, providing an exceptionally comprehensive dataset relevant to the aerosol lifecycle (Williamson et al., 2019; Brock et al.,
2019, 2021; Thompson et al., 2022; Wofsy et al., 2021). For example, the ability to measure aerosol size distributions from
below 10 nm to over 1 $\mu$m, quantify SO$_2$ at levels below 10 parts per trillion by volume (pptv), and detect hydroxyl radicals sets
the ATom observations apart from most other aircraft campaigns. Conducted across four seasons (2016-2018), ATom flights
spanning the Pacific and Atlantic Oceans from the Arctic to the Antarctic, sampling continuously both vertically (0.2 - 12
km) and latitudinally over vast remote marine regions. This makes ATom uniquely suited for evaluating ESMs simulations of
aerosol processes, especially in pristine marine environments where aerosol-cloud interactions are most sensitive and forcing
uncertainties are largest (Carslaw et al., 2013).

ATom observations have been widely used to evaluate aerosol-related processes in global models. For instance, Koenig et al.
(2020) compared CAM-Chem simulations of iodine species with measurements. Froyd et al. (2022) evaluated the performance
of CESM/CARMA and GEOS/GOCART in simulating dust aerosols and examined the influence of dust on cirrus cloud
formation. Williamson et al. (2019) compared aerosol number size distributions from ATom with outputs from GEOS-Chem,
CAM5, and CESM. Yu et al. (2019) assessed the effectiveness of aerosol removal by convective systems in CESM. Hodzic
et al. (2020) evaluated the performance of GEOS5, ECHAM6-HAM, CESM, and GEOS-Chem in simulating aerosol chemical
composition, with a particular focus on organic aerosols. Nault et al. (2021) analysed the simulated inorganic aerosol acidity
in CCSM4, GISS, TM4, GEOS-Chem, GEOS5, and AM4.1 using ATom data. More recently, Gao et al. (2022) provided a
detailed comparison of GEOS-Chem simulations with ATom measurements across multiple aerosol species, including sulfate,
organics, black carbon, nitrate, and ammonium. Bian et al. (2024) evaluated the performance of AeroCom models in simulating
the sulfur cycle over the marine atmosphere. Together, these studies demonstrated the value of ATom data in identifying model
biases and highlight the importance of accurately representing aerosol sources, transformations, and removal processes to
improve simulations of global aerosol lifecycle.

In this study, we leverage the ATom dataset to evaluate the performance of atmosphere component of the UK Earth System
Model version 1.1 (UKESM1.1) (Mulcahy et al., 2023) in simulating key aspects of the aerosol lifecycle. While an earlier
version (UKESM1.0) was previously evaluated against ATom (Ranjithkumar et al., 2021), that study focused on only three
variables related to airborne aerosol formation (SO$_2$, total aerosol number, condensation sink). Our evaluation is significantly
more comprehensive, encompassing precursor vapours (DMS, SO$_2$, NH$_3$) and oxidants (OH and ozone, O$_3$); aerosol number
concentrations across nucleation, Aitken, accumulation, and coarse modes; aerosol chemical composition (sulfate, organic,





ammonium, nitrate, seasalt); and environmental conditions (temperature [T], relative humidity [RH] and condensation sink
[CS]). Such a multi-faceted evaluation is crucial, as tuning models to match a limited subset of variables can unconsciously
introduce biases in other areas, leading to equifinality without improving overall predictive skill (Lee et al., 2016).

Furthermore, beyond evaluating the default model configuration, we implement and test several major updated or alternative
process representations within UKESM1.1. These include: (1) a new $H_2SO_4$-$NH_3$ nucleation scheme (Dunne et al., 2016)
with the ammonium nitrate scheme (Jones et al., 2021), reflecting its proposed importance in pristine environments (Kirkby
et al., 2011; Dunne et al., 2016); (2) a condensation scheme for methanesulfonic acid (MSA), an important DMS oxidation
product involved in aerosol growth (Beck et al., 2021) which is currently ignored in UKESM1.1; (3) the incorporation of an
isoprene secondary organic aerosol (SOA) formation scheme, for isoprene's significance for organic aerosol mass (Weber et al.,
2021; Tsigaridis et al., 2014); and (4) testing the impact of using a traditional DMS climatology (Lana et al., 2011) and a new
chemistry scheme, CRI-Strat2 of UKESM1.1 (Weber et al., 2021; Archer-Nicholls et al., 2021).

It is important to emphasise that while many model evaluation studies focus on sensitivity tests to improve the model
performance metrics, the primary focus of this study is improving the aerosol microphysical processes in UKESM1.1. We
prioritise implementing experimentally and observationally verified processes over optimising agreement with observations
through existing parameter tuning. We aim to identify key discrepancies between the model (in its default and modified
configurations) and the ATom observations, analysing these differences from a process-understanding perspective. Finally, we
will discuss the implications of these findings for future model development, integrating insights from laboratory studies, field
observations, and model simulations to advance our mechanistic understanding of the aerosol lifecycle and its climate role -
ensuring improvements are driven by enhanced physical representation.

## 2 Methods

### 2.1 UKESM description

This study utilises the UKESM1.1 in its atmosphere-only configuration (Sellar et al., 2019; Mulcahy et al., 2020, 2023),
wherein sea surface temperatures and sea ice are prescribed from Reynolds' database (Reynolds et al., 2007). The land surface
model (Joint UK Land Environment Simulator, JULES) is run simultaneously, but without dynamic vegetation. As a result,
land surface temperature, soil moisture, and heat and moisture fluxes to the atmosphere are simulated rather than prescribed.

UKESM1.1 builds upon the HadGEM3-GC3.1 global coupled atmosphere-ocean-ice climate model (Williams et al., 2018)
and incorporates additional Earth system components, including representations of carbon and nitrogen cycles, land use change,
ocean biogeochemistry, and a unified troposphere-stratosphere chemistry and modal aerosol scheme. The simulations in this
study employ a horizontal resolution of $1.875°$ longitude $\times$ $1.25°$ latitude, corresponding to approximately 135 km. Vertically,
the model utilises 85 levels extending up to 85 km from the Earth's surface, with 50 levels concentrated between 0 and 18 km,
which are the primary region of focus for this study.



The model simulations are nudged using wind and temperature fields from the ERA5 reanalysis (Telford et al., 2008; Dee et al., 2011) corresponding to the period of ATom observation. This nudging aims to reproduce the specific meteorological conditions at the time and location of the measurements, thereby reducing model biases often present in free-running model configurations (Kipling et al., 2013). Since the temporal resolution of the ERA5 reanalysis is 6 hours, the relaxation time
constant for the nudged simulations is set to 6 hours. Nudging is applied vertically between model levels 12 and 80.

## 2.2    United Kingdom Chemistry and Aerosol model (UKCA)

Atmospheric chemistry and aerosols are simulated using the UKCA model (version UM13.0), which is fully coupled to UKESM1.1 for handling tracer transport (O'Connor et al., 2014; Archibald et al., 2020). UKCA utilises emission datasets consistent with the CMIP6, incorporating anthropogenic (Hoesly et al., 2018), biomass-burning (Van Marle et al., 2017), and
biogenic sources (Guenther et al., 2012). As the CMIP6 emission datasets utilised in this study extend only up to 2014, emissions for the years after 2015 are prescribed using the Shared Socioeconomic Pathway (SSP) SSP3-7.0 scenario. Some biogenic emissions, including isoprene, terpenes, methanol, and acetone, are simulated using interactive emission schemes.

The aerosol scheme within UKCA is largely based on the Global Model of Aerosol Processes (GLOMAP)-mode (Mann et al., 2010; Mulcahy et al., 2020). GLOMAP employs a two-moment (tracking both number and mass) pseudo-modal ap-
proach to simulate the global distribution of sulfate, black carbon, organic matter, and seasalt aerosols, whereas mineral dust is simulated using the CLASSIC sectional dust scheme (Woodward, 2001). Nitrate and ammonium aerosols are not included in the standard UKESM1.1 configuration; however, a scheme incorporating these species has recently been developed (Jones et al., 2021) and is included in some of the simulations we present. GLOMAP simulates comprehensive aerosol microphysical processes, including NPF, condensation of vapours onto existing aerosols, aerosol coagulation, dry deposition, wet scaveng-
ing, and cloud processing (Mann et al., 2010). The aerosol number size distribution is represented by four modes: nucleation (geometric mean dry diameter, $\bar{d} < 10$ nm), Aitken ($10 < \bar{d} < 100$ nm), accumulation ($100 < \bar{d} < 500$ nm), and coarse ($500 < \bar{d} < 10{,}000$ nm). In the model version used in this study, all four modes contain soluble components. Additionally, there is an insoluble Aitken mode ($100 < \bar{d} < 500$ nm) composed solely of organic matter and black carbon. At each timestep, a fraction of insoluble Aitken-mode particles is transferred to the soluble Aitken mode. This fraction is proportional to the condensation
rate of soluble material (e.g. sulfate or organics) and is scaled such that the accumulation of ten monolayers of soluble material on a fraction of insoluble aerosols would result in the conversion to the soluble mode.

Since comprehensive descriptions of the atmospheric chemistry and aerosol formation mechanisms in UKCA have been previously published (Mann et al., 2010; O'Connor et al., 2014; Archibald et al., 2020; Mulcahy et al., 2020), we focus here only on the key processes and parameterisations directly relevant to the analysis presented in this study.




### 2.2.1 Primary aerosols

Primary aerosols are aerosols directly emitted from various sources. In UKESM1.1, these include mineral dust, seasalt, black carbon, organic matter, and sulfate. Mineral dust is simulated using a scheme with six size bins ranging from 0.06 to 60 $\mu$m in diameter. The dust within each bin is treated independently and assigned a density of 2.65 kg m$^{-3}$ (Woodward, 2001).

Seasalt emissions are calculated using the bin-resolved parameterisation of Gong (2003). The emitted mass and number are distributed between the soluble accumulation and coarse modes, depending on whether the centre diameter of the source bin is below or above the diameter threshold separating these modes (approximately 500 nm). A density of 2.165 kg m$^{-3}$ is assumed for seasalt in UKESM1.1.

    Primary carbonaceous aerosol emissions include black carbon and organic matter originating from both anthropogenic

sources (biofuel and fossil fuel combustion) and biomass burning processes. Aerosols emitted from biomass/biofuel sources are assigned a geometric mean diameter of 150 nm, while those from fossil fuel sources are assigned 60 nm; both emission types assume standard deviation of 1.59 (Stier et al., 2005; Mann et al., 2010). Notably, although the assigned diameter for biomass/biofuel aerosol emissions (150 nm) exceed the typical upper size limit of the Aitken mode (100 nm), these aerosols are nevertheless emitted into the model's insoluble Aitken mode. Anthropogenic emissions are released into the lowest model

layer, whereas biomass burning emissions are distributed vertically between the surface and approximately 6 km above ground level.

    Primary marine organic aerosols represent another primary aerosol source which was recently implemented in UKESM (Mulcahy et al., 2020). These organic aerosols are thought to be emitted as components of organic-enriched sea spray (Rinaldi et al., 2010), and their emissions show a high correlation with marine biological activity, often indicated by chlorophyll con-

centrations (Rinaldi et al., 2013). Consequently, UKESM1.1 adopts the parameterisation of Gantt et al. (2012), which relates emissions to both wind speed and biological activity represented by surface chlorophyll concentration. The calculated organic mass emission flux is partitioned, with 25 % attributed to the soluble Aitken mode and 75 % to the insoluble Aitken mode. An emission diameter of 160 nm is assumed for both fractions, based on experimental and observational constraints (O'Dowd et al., 2004; Prather et al., 2013).

Consistent with Mann et al. (2010), the model assumes that 2.5 % of anthropogenic SO$_2$ emissions by mass are directly emitted as primary sulfate aerosols. This primary sulfate mass is distributed with an initial size distribution specified by Stier et al. (2005): 50 % is allocated to the accumulation mode (assuming $\bar{d} = 150$ nm) and 50 % to the coarse mode (assuming $\bar{d} = 1500$ nm).

### 2.2.2 Sulfur sources and chemistry

Sulfate aerosols form through the oxidation of SO$_2$, either via gas-phase reactions that produce H$_2$SO$_4$ and trigger NPF and growth, or through multiphase oxidation by hydrogen peroxide (H$_2$O$_2$) and O$_3$ dissolved in cloud liquid water, which contributes to aerosol mass only. The gas-phase oxidation of SO$_2$ produces gaseous H$_2$SO$_4$, which drives NPF and growth in the



model. In the aqueous phase, $SO_2$ dissolves into cloud droplets, where it undergoes oxidation by dissolved $H_2O_2$ and $O_3$ to
form sulfate. The model does not include an explicit representation of multi-phase chemical species; instead, the sulfate formed
through these pathways is treated as a direct mass flux contributing to the accumulation and coarse aerosol modes. Finally, because certain removal processes (e.g. precipitation removal) associated with aqueous sulfate formation are not represented in
the model, a 25 % reduction factor is applied to the calculated aqueous sulfate formation mass flux (Mulcahy et al., 2020).

In UKCA, $SO_2$ originates from both the oxidation of DMS and direct anthropogenic emissions. Anthropogenic emissions
of $SO_2$ are taken from the Community Emissions Data System (CEDS) inventory (Hoesly et al., 2018). $SO_2$ emitted from
both the energy and industrial sectors is released into the model's surface layer, consistent with the treatment of other trace
gas emissions in UKCA. Additionally, natural $SO_2$ emission from continuously degassing volcanoes is prescribed using the
climatology developed by Dentener et al. (2006). A major revision of the $SO_2$ dry deposition scheme, as described by Hardacre
et al. (2021), was implemented in UKESM1.1. This update, along with several additional bug fixes and model improvements
detailed in Mulcahy et al. (2023), contributed to reduced surface $SO_2$ concentrations in UKESM1.1 compared to the earlier
UKESM1 version evaluated by Ranjithkumar et al. (2021).

Marine DMS emissions in the model are calculated using an interactive scheme. The ocean biogeochemistry component,
incorporating the Model of Ecosystem Dynamics, nutrient Utilisation, Sequestration and Acidification (MEDUSA) module,
simulates the DMS concentration in the surface ocean. This follows the formulation modified from Anderson et al. (2001)
to ensure energy balance within the coupled system (Sellar et al., 2019). Additionally, terrestrial DMS emissions are also
included, based on an earlier climatology (Spiro et al., 1992).

The representation of DMS chemistry within UKCA is simplified, a common necessity in large-scale global models. The
primary oxidation pathway for DMS represented in the model is the reaction with OH radicals, which is the dominant atmospheric sink for DMS. Additionally, DMS is also oxidised by nitrate ($NO_3$) radicals and atomic oxygen ($O(^3P)$), although
the latter pathway is generally negligible in the troposphere. The default chemistry scheme employed in UKCA is the StratTrop scheme (Archibald et al., 2020), but the specific reactions of DMS oxidation pathways differ between UKESM1 and
UKESM1.1 (Table 1). The reaction of DMS with OH proceeds via two channels: addition and abstraction. In both UKESM
versions, the abstraction channel of OH oxidation, along with DMS oxidation by $NO_3$, leads directly to the formation of $SO_2$.
While Mulcahy et al. (2023) suggested that the UKESM1.1 included the reaction $DMS + O(^3P) \rightarrow SO_2$, it is not included
in the version of UKESM1.1 used in this study. The addition channel in UKESM1.1 differs from that in UKESM1; the latter
neglects the formation of dimethyl sulfoxide (DMSO). However, DMSO is recognised as an important intermediate species
subject to atmospheric transport and deposition and it is a precursor of MSA. Consequently, DMS oxidation by OH directly
produces MSA in UKESM1.0, partially accounting for the neglected DMSO pathway. In UKESM1.1, the addition channel
was modified to produce a mixture of $SO_2$ and DMSO, with molar yields of 0.6 and 0.4, respectively. Subsequent DMSO
oxidation by OH in UKESM1.1 further produces $SO_2$ and MSA, with yields of 60 % and 40 %, respectively (Table 1).



**Table 1.** Comparison of DMS oxidation pathways across different chemistry schemes in UKESM versions. The table shows the key DMS oxidation reactions implemented in UKESM1.0 (Strat-Trop), UKESM1.1 (Strat-Trop), and UKESM1.1 (CRI-Strat2) chemistry schemes. UKESM1.1 incorporates dimethyl sulfoxide (DMSO) as an intermediate species with simplified oxidation pathways, while CRI-Strat2 includes a more detailed representation with methylthiomethylperoxy radical (MTMP) and methylsulfinic acid (MSIA) intermediates leading to both $SO_2$ and MSA formation. The numerical coefficients indicate stoichiometric yields for branching reactions. We note that while Mulcahy et al. (2023) suggested that the UKESM1.1 included $DMS + O(^3P) \rightarrow SO_2$, it is not included in the version of UKESM1.1 used in this study.

| UKESM1(Strat-Trop) | UKESM1.1(Strat-Trop) | UKESM1.1(CRI-Strat2) |
|---|---|---|
| $DMS + OH \rightarrow SO_2$ | $DMS + OH \rightarrow SO_2$ | $DMS + OH \rightarrow MTMP + H_2O$ |
| $DMS + OH \rightarrow SO_2 + MSA$ | $DMS + OH \rightarrow 0.6SO_2 + 0.4DMSO$ | $DMS + NO_3 \rightarrow MTMP + HNO_3$ |
| $DMS + NO_3 \rightarrow SO_2$ | $DMS + NO_3 \rightarrow SO_2$ | $DMS + OH \rightarrow DMSO + HO_2$ |
| $DMS + O(^3P) \rightarrow SO_2$ | $DMSO + OH \rightarrow 0.6SO_2 + 0.4MSA$ | $MTMP + NO \rightarrow HCHO + CH_3S + NO_2$ |
| | | $MTMP + MTMP \rightarrow 2HCHO + 2CH_3S$ |
| | | $CH_3S + O_3 \rightarrow CH_3SO$ |
| | | $CH_3S + NO_2 \rightarrow CH_3SO + NO$ |
| | | $CH_3SO + NO_2 \rightarrow CH_3SO_2 + NO$ |
| | | $CH_3SO + NO_2 \rightarrow SO_2 + CH_3O_2 + NO$ |
| | | $CH_3SO + O_3 \rightarrow CH_3SO_2$ |
| | | $DMSO + OH \rightarrow MSIA + CH_3O_2$ |
| | | $MSIA + OH \rightarrow CH_3SO_2 + H_2O$ |
| | | $MSIA + OH \rightarrow MSA + HO_2 + H_2O$ |
| | | $MSIA + NO_3 \rightarrow CH_3SO_2 + HNO_3$ |
| | | $CH_3SO_2 \rightarrow CH_3O_2 + SO_2$ |
| | | $CH_3SO_2 + O_3 \rightarrow CH_3SO_3$ |
| | | $CH_3SO_2 + NO_2 \rightarrow CH_3SO_3 + NO$ |
| | | $CH_3SO_3 + HO_2 \rightarrow MSA$ |
| | | $CH_3SO_3 \rightarrow CH_3O_2 + H_2SO_4$ |

### 2.2.3 Airborne formation of aerosols

Several key airborne aerosol production processes are simulated by default in UKCA. First, new particles are formed via the binary $H_2SO_4 - H_2O$ nucleation scheme, producing aerosols in the smallest (nucleation) mode. Second, aerosols grow via the condensation of $H_2SO_4$ and organic oxidation products onto existing aerosol surfaces. In the meantime, coagulation between aerosol particles reduces the number concentration in smaller modes (e.g. the nucleation mode) while contributing to aerosol growth in larger modes (e.g. the Aitken and accumulation modes). When aerosols within a specific mode grow beyond the upper size threshold for that mode, they are transferred to the next largest mode via mode merging (Mann et al., 2010).




The default binary $H_2SO_4-H_2O$ aerosol nucleation mechanism follows the theoretical parameterisation described by
Vehkamäki et al. (2002), which is most effective at producing new particles in the free troposphere. Mechanisms for NPF
specifically within the planetary boundary layer are not explicitly included in the default model configuration. However, the
model includes an option to use a boundary layer nucleation parameterisation based on cluster activation theory, wherein the
nucleation rate exhibits a power-law dependence on the $H_2SO_4$ concentration (Kulmala et al., 2006). Additionally, a multi-
component nucleation scheme described by Metzger et al. (2010), which considers the roles of both $H_2SO_4$ and organic
vapours, is also available as an option within the model. It should be noted that these optional boundary layer nucleation
schemes do not quantitatively represent the state-of-the-science atmospheric nucleation mechanisms and are therefore not
utilised in this study (Kirkby et al., 2023).

Condensable organic oxidation products, which contribute to aerosol growth, are represented in the model as originating
from the oxidation of monoterpenes, assuming a lumped mass yield of 13 % (Mann et al., 2010). However, to account for SOA
production from other sources (such as isoprene oxidation) and to compensate for the absence of explicit anthropogenic and
marine volatile organic compound (VOC) emissions in this model configuration, this lumped monoterpene oxidation yield is
scaled by a factor of two.

Besides nucleation, vapour condensation, and coagulation, other aerosol microphysical processes simulated in the model
impact the aerosol lifecycle, including dry deposition, wet scavenging, and the ageing of insoluble aerosols (representing
processes that increase aerosol solubility). Readers are referred to Mann et al. (2010) for the detailed formulation of these
microphysical processes.

## 2.3  Model development

A major hurdle in climate simulations involving aerosols and their interactions with clouds and climate is the often-poor
representation of precursor chemistry (e.g. sulfur chemistry) and aerosol nucleation mechanisms. For example, the theoretical
binary $H_2SO_4-H_2O$ nucleation mechanism adopted by UKESM1.1 dates back to around the turn of the century (Kulmala
et al., 1998; Vehkamäki et al., 2002), and this mechanism is widely recognised as mechanistically insufficient to fully explain
atmospheric aerosol formation (Kirkby et al., 2011, 2023). Furthermore, recent advances in understanding DMS oxidation
mechanisms point to insufficient representation of these pathways in the default UKESM1.1 for accurately simulating marine
sulfur chemistry and subsequent aerosol formation (Cala et al., 2023).

In addition to employing potentially outdated mechanisms, the default UKESM1.1 configuration also lacks representations
of several processes known to be important for aerosol formation and growth. For instance, the model lacks representation of
ion-induced nucleation processes, which are recognised as a globally significant source of aerosols (Kirkby et al., 2011; Dunne
et al., 2016). Additionally, the model treats MSA as an inert tracer, meaning it is not subject to deposition, or condensation
processes, despite its demonstrated importance in aerosol growth (Beck et al., 2021). The model also lacks a representation of
NPF processes involving isoprene oxidation products, which has recently been identified as important in the tropical UT (Shen
et al., 2024; Curtius et al., 2024). Other key omissions include, for example, the lack of representation of iodine chemistry



and the associated iodine oxoacid aerosol formation mechanisms, processes shown to have a global impact on marine aerosol formation (He et al., 2021b, 2023), and alkaline molecules such as ammonia ($NH_3$), which are known to enhance aerosol

nucleation (Kirkby et al., 2011).

To address some of these key deficiencies, this study implements several process improvements, focusing primarily on incorporating the $H_2SO_4-NH_3$ nucleation mechanism, which includes both the ion-induced and neutral nucleation channels, and a new, preliminary MSA condensation scheme based on recent observational findings. $H_2SO_4-NH_3$ aerosol nucleation has been shown to be an important global process (Kirkby et al., 2011; Dunne et al., 2016). The implemented $H_2SO_4-NH_3$

nucleation parameterisation is one of the few derived directly from experiments utilising instrumentation designed to minimise the influence of organic vapours and other strong alkaline molecules (Kirkby et al., 2011; Almeida et al., 2013). The MSA condensation scheme is based on the observational work of Beck et al. (2021), which demonstrated that MSA can condense effectively onto pre-existing aerosols, contributing significantly to their early growth. As an initial proof-of-concept implementation, this MSA condensation scheme assumes irreversible condensation onto pre-existing aerosols (analogous to the model's

treatment of $H_2SO_4$ and oxidised organics).

In addition to these new developments, we also test the impact of using the alternative CRI-Strat2 chemistry scheme (Weber et al., 2021; Archer-Nicholls et al., 2021), incorporate the recently developed ammonium nitrate scheme (Jones et al., 2021), and evaluate the effect of replacing the interactive DMS emission scheme with a DMS climatology (Lana et al., 2011) in the model. A full list of the simulations performed is provided in Table 2, which details the key differences between the model

configurations used in this study.

### 2.3.1 Development of $H_2SO_4-NH_3$ aerosol nucleation mechanism

The implemented $H_2SO_4-NH_3$ nucleation scheme is based on the parameterisation developed by Dunne et al. (2016), which was derived from measurements performed in the CERN CLOUD (Cosmics Leaving OUtdoor Droplets) experiments. A key feature of the CLOUD experiments, compared with previous laboratory studies, is the extremely high standard of cleanliness.

This allows the nucleation capability of individual vapours, and mixtures thereof, to be studied independently with minimal contamination. Taking advantage of this capability, experiments were performed to separately evaluate the influence of $H_2SO_4$, $NH_3$, $H_2O$, and atmospheric ions on the aerosol nucleation rate. Readers are referred to the original paper for detailed formulations and parameter values (Dunne et al., 2016).

Experiments reported in (Dunne et al., 2016) identified four distinct aerosol nucleation regimes, differentiated by the con-

centrations of $NH_3$ and the presence of atmospheric ions. The first regime is termed binary neutral nucleation of $H_2SO_4-H_2O$; this represents the experimental version of the theoretical nucleation scheme used by default in UKESM1.1 (Kulmala et al., 1998; Vehkamäki et al., 2002). This regime's rate depends only on the concentrations of $H_2SO_4$ and $H_2O$. The second regime is termed binary ion-induced nucleation of $H_2SO_4-H_2O$, describing nucleation in the presence of atmospheric ions. Consequently, the nucleation rate of this regime exhibits an additional dependence on the concentration of atmospheric ions. The

third regime is ternary neutral nucleation of $H_2SO_4-NH_3(-H_2O)$, which depends on the concentrations of $H_2SO_4$, $NH_3$, and $H_2O$. As $H_2O$ participation is fundamental to atmospheric nucleation, it is often implicitly assumed and omitted from the



**Table 2.** Summary of model simulations and configurations used in this study. The table shows the key differences between simulations, including DMS emission schemes (interactive vs. climatology), atmospheric chemistry schemes (Strat-Trop vs. CRI-Strat2), aerosol nucleation mechanisms ($H_2SO_4-H_2O$ vs. $H_2SO_4-NH_3$), inclusion of ammonium nitrate chemistry, methanesulfonic acid (MSA) condensation, and secondary organic aerosol (SOA) formation pathways. MTSOA refers to monoterpene-derived SOA mass yield, while IPSOA refers to isoprene-derived SOA mass yield. The SA-NH$_3$ (benchmark) simulation represents the primary configuration with newly implemented $H_2SO_4-NH_3$ nucleation scheme and ammonium nitrate scheme used for comparison with other model implementations. Simulation abbreviations (e.g. SA-NH$_3$-IPSOA) are used throughout the manuscript, where SA denotes sulfuric acid to distinguish from the chemical formula notation (e.g. $H_2SO_4-NH_3$) used for nucleation mechanisms.

| Simulations | DMS | Chemistry | Nucleation | Nitrate | MSA condensation | MTSOA | IPSOA |
|---|---|---|---|---|---|---|---|
| SA-H$_2$O(default) | Interactive | Strat-Trop | H$_2$SO$_4$-H$_2$O | N/A | N/A | 26% | N/A |
| SA-NH$_3$-noNit | Interactive | Strat-Trop | H$_2$SO$_4$-NH$_3$ | N/A | N/A | 26% | N/A |
| SA-NH$_3$-slow | Interactive | Strat-Trop | H$_2$SO$_4$-NH$_3$ | Yes (slow) | N/A | 26% | N/A |
| SA-NH$_3$-Lana | Climatology | Strat-Trop | H$_2$SO$_4$-NH$_3$ | Yes | N/A | 26% | N/A |
| SA-NH$_3$(benchmark) | Interactive | Strat-Trop | H$_2$SO$_4$-NH$_3$ | Yes | N/A | 26% | N/A |
| SA-NH$_3$-CS2 | Interactive | CRI-Strat2 | H$_2$SO$_4$-NH$_3$ | Yes | N/A | 26% | N/A |
| SA-NH$_3$-MSA | Interactive | Strat-Trop | H$_2$SO$_4$-NH$_3$ | Yes | Yes | 26% | N/A |
| SA-NH$_3$-IPSOA | Interactive | Strat-Trop | H$_2$SO$_4$-NH$_3$ | Yes | N/A | 13% | 3% |
| SA-NH$_3$-IPSOA×10 | Interactive | Strat-Trop | H$_2$SO$_4$-NH$_3$ | Yes | N/A | 13% | 30% |

nomenclature. Finally, the fourth regime is ternary ion-induced nucleation of $H_2SO_4-NH_3$, which incorporates the enhancing effect of atmospheric ions on the ternary system.

As atmospheric ion pair production rates are not explicitly simulated in UKESM1.1, they are prescribed using a climatology output from the simulations described by Gordon et al. (2017). NH$_3$ emissions in UKESM1.1 are taken from the CEDS dataset (Hoesly et al., 2018). However, simulated atmospheric NH$_3$ concentrations within the model are not well constrained. For example, the uptake of gaseous ammonia by acidic aerosols (leading to ammonium formation), which is widely acknowledged as an important NH$_3$ sink, is not represented in the default model configuration. This leads to an overestimation of gas-phase NH$_3$ concentrations in the model. Therefore, the recently developed ammonium nitrate scheme (Jones et al., 2021) is incorporated in most simulations in this study, aiming to improve the representation of atmospheric NH$_3$ budget.

The new $H_2SO_4-NH_3$ nucleation scheme is implemented in all simulations listed in Table 2, except for the control simulation using the default binary scheme, labelled $SA-H_2O$(default). The SA-NH$_3$(benchmark) simulation represents a key experiment in this study; it incorporates the ammonium nitrate scheme using parameter settings corresponding to the fast nitrate formation scheme (specifically, a nitric acid uptake coefficient $\gamma = 0.193$) as described by Jones et al. (2021). Another sensitivity simulation, SA-NH$_3$-slow, explores slower nitrate formation by utilising $\gamma = 0.001$. Additionally, the SA-NH$_3$-noNit simulation is performed using the $H_2SO_4-NH_3$ nucleation scheme but without activating the ammonium nitrate module, serving as a reference to evaluate the impact of the nitrate scheme itself on aerosol formation.





It is worth noting that the representation of the nucleation rate enhancement by $H_2O$ differs in this study compared to the original formulation in Dunne et al. (2016). In both studies, the impact of $H_2O$ is represented as a relative humidity dependent multiplier ($K_{RH}$) applied to the nucleation rate calculated based on $H_2SO_4$, $NH_3$, and atmospheric ion concentrations. The original formula reads $K_{RH} = 1 + c_1(RH - 0.38) + c_2(RH - 0.38)^3(T - 208)^2$, where $c_1 = 1.5 \pm 1.3$, $c_2 = 0.045 \pm 0.003$, RH is the relative humidity expressed as a fraction, and T is the temperature in Kelvin. This formulation uses 38 % RH (RH = 0.38) as the reference condition, reflecting the humidity level at which most of the underlying experiments were performed. Unfortunately, this formulation was derived from a relatively limited number of experiments conducted at other humidity levels, and its associated uncertainty is large (Dunne et al., 2016). We find that this RH enhancement factor likely exaggerates the humidity effect at both high (> 90 %) and low (< 20 %) relative humidities, potentially distorting the calculated nucleation rate under these conditions. Therefore, a more conservative formulation for the RH dependence is adopted in this study: $K_{RH} = (RH/0.38) \times [1 + 0.02 \times (RH - 0.38) \times (T - 208)^{1.2}]$.

### 2.3.2 Development of MSA condensation scheme

In the default UKESM1.1 configuration (Mulcahy et al., 2023), MSA is treated as an inert tracer. This treatment causes MSA to accumulate indefinitely in the model, leading to unrealistically high concentrations. To address this limitation, a scheme of MSA condensation onto pre-existing aerosols, as well as its wet and dry deposition, are implemented in the model in this study.

The wet deposition of MSA is treated analogously to other soluble gas-phase species in the model, following the formulation described by Giannakopoulos et al. (1999). Consistent with recommendations by Barnes et al. (2006) and Cala et al. (2023), an effective Henry's law constant of $10^9$ M atm$^{-1}$ is adopted for MSA. The dry deposition rate of MSA is assumed to be the same as that calculated for gaseous $H_2SO_4$, for the similarity in their molecular sizes and, consequently, their diffusivities in air.

Finally, the MSA condensation scheme implemented assumes irreversible condensation onto pre-existing aerosol particles, at the same rate as $H_2SO_4$. This assumption is based on the comparable molecular weights (MSA = 96 g mol$^{-1}$; $H_2SO_4$ = 98 g mol$^{-1}$) and bulk densities (MSA = 1.5 g cm$^{-3}$; $H_2SO_4$ = 1.8 g cm$^{-3}$) of the two species. This treatment is supported by recent Arctic observations which suggest that MSA can condense effectively onto pre-existing aerosols, potentially at a rate comparable to that of $H_2SO_4$ (Beck et al., 2021). The MSA aerosol mass is currently merged with the existing sulfate aerosol mass in the model for simplicity. The MSA condensation scheme is added to the SA-$NH_3$ (benchmark), hereafter referred to as SA-$NH_3$-MSA.

However, it must be emphasised that this MSA condensation scheme is preliminary and has not yet been fully validated against comprehensive experimental data. For instance, a potential humidity dependence of MSA partitioning to the aerosol phase has been implicated in aircraft measurements (Mauldin et al., 1999). Other modelling studies have adopted volatility-dependent parameterisations for MSA condensation, assuming it to be a temperature- and humidity-dependent process (Hodshire et al., 2019). Unfortunately, there is currently limited experimental data available to rigorously validate MSA condensation rates predicted by either type of scheme; future experimental work is crucial for developing and constraining





MSA condensation parameterisations in models. Thus, the MSA condensation scheme in this study aims to provide an initial proof-of-concept implementation, allowing for the evaluation of MSA's potential impact within the current chemistry framework.

370

### 2.3.3 Coupling $H_2SO_4 - NH_3$ nucleation with a DMS emission climatology

Although the default UKESM1.1 configuration employs an interactive marine DMS emission scheme coupled to the MEDUSA module, this study also evaluates the widely used DMS climatology developed by Lana et al. (2011). This comparison is motivated by the significant uncertainties associated with modelled DMS emissions. For example, Bhatti et al. (2023) demonstrated that simulated DMS emissions in the Southern Ocean are highly sensitive to the choice of emission parameterisation. Specifically, the seawater DMS concentrations predicted by the interactive MEDUSA scheme generally exhibit limited spatial variability, although they show low values surrounding the Antarctic continent. In contrast, the Lana et al. (2011) climatology predicts distinct hotspots of high seawater DMS concentration near Antarctica. While rigorously verifying the true spatial pattern of global DMS emissions is beyond the scope of this study, the distinct differences between the interactive scheme and the climatology are substantial. This provides an opportunity to evaluate the sensitivity of simulated atmospheric chemistry and aerosol properties in response to different DMS source mechanisms. The DMS climatology is implemented in the simulation labelled SA-NH₃-Lana, which also utilises the $H_2SO_4 - NH_3$ nucleation scheme and the ammonium nitrate scheme.

### 2.3.4 Coupling $H_2SO_4 - NH_3$ nucleation with CRI-Strat2 chemistry

The coupling of the Common Representative Intermediates mechanism (CRI) with the existing UKCA stratospheric chemistry scheme (Strat) was initially carried out by Archer-Nicholls et al. (2021) and subsequently updated by Weber et al. (2021) to create the CRI-Strat2 (CS2) chemistry scheme. The development of CS2 aimed to improve the model's representation of the oxidation of non-methane volatile organic compounds and provide traceability to the CRI2.2 scheme (Jenkin et al., 2019). For instance, compared to the default Strat-Trop scheme, the CS2 scheme includes a more detailed representation of the oxidation pathways for DMS and other key biogenic VOCs, such as isoprene and monoterpenes. It is worth noting that the CS2 scheme is not only a new chemistry mechanism but also uses different emission inventories - distinct from those used in simulations with the Strat-Trop chemistry scheme. Details of the emission inventories employed in the CS2 scheme are provided in Archer-Nicholls et al. (2021); Weber et al. (2021). To better represent NH₃ levels in the marine atmosphere, we incorporated the marine NH₃ emissions from Bouwman et al. (1997), originally used in the Strat-Trop scheme, into the CS2 simulation.

The key reaction pathways of the DMS oxidation within the CS2 version used in this study are tabulated in Table 1. Similarly to other sensitivity simulations, the CS2 scheme is coupled with the $H_2SO_4 - NH_3$ nucleation scheme and the ammonium nitrate scheme, which is labelled SA-NH₃-CS2. It is important to note, however, that despite providing an improved representation compared to Strat-Trop, the key DMS oxidation pathways included in CS2 (primarily following those described by Von Glasow and Crutzen (2004)) do not reflect the latest scientific understanding (Cala et al., 2023). Numerous recent studies,





emerging from theoretical, laboratory experiments, and field observations, have provided a more comprehensive picture of DMS oxidation mechanisms (Veres et al., 2020; Shen et al., 2022; Jacob et al., 2024). However, incorporating these more up-to-date reaction pathways into the model is beyond the scope of this study. The primary purpose of this simulation is therefore to evaluate how employing a different, more complex tropospheric chemistry scheme (CS2 vs. Strat-Trop) influences the simulated atmospheric sulfur cycle, the resultant aerosol number size distributions and composition.


### 2.3.5 Coupling $H_2SO_4 - NH_3$ nucleation with isoprene secondary organic aerosol formation

The default UKESM1.1 configuration incorporates a simplified representation of SOA formation, based solely on the oxidation of monoterpenes. SOA formation from isoprene oxidation is not explicitly represented; instead, its contribution is implicitly considered by scaling the monoterpene oxidation SOA yield by a factor of two. However, Weber et al. (2022) recently im-

plemented a new scheme in the UKCA model to explicitly represent isoprene SOA formation. This new scheme treats SOA formed from isoprene oxidation as a distinct aerosol component, produced with a fixed mass yield of 3 % from the reaction of isoprene with the major atmospheric oxidants ($O_3$, $NO_3$, and OH). Other aspects of the treatment of this isoprene SOA component are assumed to be identical to those of the SOA derived from monoterpene oxidation. The choice of the 3 % yield is based on the work of Scott et al. (2014), which in turn relies on the experimental findings of Kroll et al. (2005, 2006). Con-

currently with the introduction of this explicit isoprene SOA source, the scaling factor previously applied to the monoterpene SOA yield is removed. Therefore, the mass yield for SOA formation from monoterpene oxidation reverts to its base value of 13 % in simulations using this scheme (Table 2). The simulation incorporating this explicit isoprene SOA scheme is referred to as SA-NH$_3$-IPSOA.

It should be acknowledged that representing SOA formation using fixed yields is a significant simplification, as actual

yields are known to be strongly influenced by factors such as temperature, oxidant concentrations, aerosol acidity and nitrogen oxide ($NO_x$) levels. Furthermore, the formation pathways differ: monoterpene oxidation can produce extremely low-volatility compounds that contribute effectively to SOA mass through irreversible condensation (Ehn et al., 2014), whereas isoprene SOA formation is thought to be dominated by the reactive uptake of gas-phase oxidation products, such as isoprene epoxydiols (IEPOX) (Paulot et al., 2009). This reactive uptake process is significantly affected by factors like aerosol acidity; for instance,

yields as high as 28.6 % have been observed under low-$NO_x$ conditions onto acidified sulfate seed aerosol (Surratt et al., 2010). Additionally, many previous laboratory experiments on isoprene SOA yields were conducted at room temperatures. Given that lower temperatures reduce the volatility of organic vapours, the effective yield of isoprene SOA formation is expected to be higher under colder atmospheric conditions. This expectation is supported by the recent work of Shen et al. (2024), which revealed that isoprene oxidation products can initiate both aerosol nucleation and subsequent growth at low temperatures, im-

plying a significantly higher potential SOA yield under such conditions. Therefore, to explore the sensitivity to this uncertainty, we also perform a simulation where the mass yield for the explicit isoprene SOA formation scheme is increased tenfold (to 30 %). This simulation is referred to as SA-NH$_3$-IPSOA×10.



## 2.4 ATom airborne campaign

The ATom airborne campaign aimed at investigating the composition of the atmosphere over the remote Pacific and Atlantic Oceans (Thompson et al., 2022). Measurements were carried out using the NASA DC-8 aircraft from 2016 to 2018 across four major deployments, roughly corresponding to the four seasons: spring (ATom-4, April-May 2018), summer (ATom-1, July-August 2016), autumn (ATom-3, September-October 2017), and winter (ATom-2, January-February 2017). The measurements covered a wide latitudinal range from the Arctic to the Antarctic (approximately 84 °N to 86 °S) and extended vertically from

near the surface to the tropopause (approximately 0.2 to 12 km), primarily over remote ocean regions. The aerosol number size distribution data used in this study are from version 2 of the dataset by Brock et al. (2022), while the trace gas data are from Wofsy et al. (2021).

One of the key scientific objectives of the ATom campaign is to understand the distribution of aerosols and the precursor vapours contributing to aerosol formation and growth. Consequently, the ATom campaign boasted a comprehensive suite of

instruments measuring the atmospheric constituents relevant to the full aerosol lifecycle, from precursor vapours to aerosol number size distribution and chemical composition. This rich dataset provides a unique opportunity to evaluate the performance of the UKESM1.1 model simulations presented in this study.

The aerosol number size distribution of dry aerosols, spanning the diameter range from 2.7 nm to 4.8 $\mu$m, was obtained by merging measurements from several instruments. Firstly, the nucleation-mode aerosol size spectrometer (NMASS) system

features a set of five condensation particle counters (CPCs) operating in parallel at different cut-off diameters (Williamson et al., 2018). Two sets of NMASS instruments were deployed on the DC-8 aircraft: the first was used across all four ATom campaigns, while the second was added for ATom-2, -3, and -4. The CPCs of the first NMASS had lower cut-off diameters set at 3.2, 8.3, 14, 27, and 59 nm, while the second NMASS used cut-offs of 5.2, 6.9, 11, 20, and 38 nm, resulting in merged data across 10 size bins below approximately 60 nm. Aerosols in the diameter range 63 nm to 1000 nm were measured by

an ultra-high-sensitivity aerosol size spectrometer (UHSAS) (Kupc et al., 2018). For larger aerosols from 120 nm to 10 $\mu$m, a laser aerosol spectrometer (LAS, model 3340, TSI Inc.) was employed. However, the effective upper detection size limit for the LAS was restricted to 4.8 $\mu$m due to the size cut by the aircraft inlet system (Brock et al., 2019). Furthermore, due to an instrument malfunction during ATom-2, LAS data from that deployment were only used for the 0.97 $\mu$m to 4.8 $\mu$m range, after the application of correction factors (Brock et al., 2019). It is important to note that aerosol size distribution data exclude

periods when the aircraft was within clouds (Brock et al., 2019).

Regarding precursor gases, the $SO_2$ data from ATom-1, -2, and -3 had insufficient sensitivity (detection limit > 100 pptv) for typical remote marine conditions (tens of pptv). Therefore, only the $SO_2$ data from ATom-4 are used in this evaluation. The instrument used during the ATom-4 campaign was a laser-induced fluorescence (LIF) instrument capable of detecting $SO_2$ down to approximately 2 pptv, even at pressures as low as 35 hPa, making it ideal for the ATom measurement requirements

(Rollins et al., 2016). OH concentrations were measured using the Penn State Airborne Tropospheric Hydrogen Oxides Sensor (ATHOS), which reported detection limits of approximately $4.5 \times 10^5$ cm$^{-3}$ near the surface and $1.5 \times 10^5$ cm$^{-3}$ at 10 km alti-



tude. O$_3$ concentrations were measured by the National Oceanic and Atmospheric Administration (NOAA) NO$_y$O$_3$ instrument (Pollack et al., 2010).

Measurements of non-refractory submicron aerosol chemical composition were provided by the University of Colorado
high-resolution time-of-flight aerosol mass spectrometer (AMS) (Hodzic et al., 2020; Guo et al., 2021). For consistency with the model's aerosol scheme, the evaluated AMS composition data were limited to sulfate, ammonium, nitrate, organic matter, and seasalt components. The atmospheric NH$_3$ concentration data used in this study were not directly measured but were derived from the AMS aerosol acidity measurements provided by Nault et al. (2021). Therefore, the estimation of gas-phase NH$_3$ is likely subject to greater uncertainty than that of aerosol mass measurements.

Unless otherwise noted, reported observational and model data for aerosol number concentrations, vapour condensation sink, and aerosol mass concentrations used in this study have been converted to standard temperature and pressure (STP: 273.15 K, 1000 hPa). To enable direct comparisons between the model output and observations, aerosol number size distributions from ATom were recalculated into consistent size modes: nucleation mode (< 10 nm in diameter), Aitken mode (10 nm - 100 nm), accumulation mode (100 nm - 1 $\mu$m), and coarse mode (1 $\mu$m - 10 $\mu$m).


## 2.5 Evaluation of UKESM1.1 using ATom observations

For comparison with ATom observations, model outputs are retrieved as instantaneous values at a high temporal resolution of one hour. This high frequency is intended to minimise sampling bias from the model. However, due to the substantial disk space required for these high-resolution outputs (roughly 25 Gigabytes per day), model data are saved only for the specific
dates corresponding to ATom flights for subsequent offline analysis.

Since this study focuses on remote marine environments, ATom observations over the continental United States and Canada are excluded from the analysis. Most of the ATom observations used in this study have 1-minute time resolution, including aerosol number size distribution (Brock et al., 2019, 2021), environmental conditions (T, RH, CS), particle composition and NH$_3$ (Nault et al., 2021). DMS, O$_3$, OH and SO$_2$ used in this study have time resolution between 120 - 200 seconds (Wofsy
et al., 2021). For consistency, the ATom data are sub-sampled to a fixed 5-minute interval by selecting the nearest data point within each 5-minute window. This interval is selected based on a combined assessment of the temporal resolutions of various parameters measured during the ATom campaign, the model's output resolution, and the objective of obtaining multiple data points within each model grid box (assuming a NASA DC-8 average speed of 833 km h$^{-1}$, a 5-minute flight segment covers roughly 70 km). Since the aerosol number size distribution is the primary dataset that underpins the entire analysis, all other
parameters are included only when corresponding aerosol data are available. This ensures that the temporal and spatial coverage of the supporting data aligns with that of the aerosol measurements. Following data preparation, the model outputs are interpolated onto the four-dimensional grid (longitude, latitude, altitude, and time) of the observational data using the In-Situ Observations Simulator (Russo et al., 2025). It is important to note that this study makes frequent use of model-to-ATom ratios, as well as their logarithmic form ($\log_{10}(\text{model}/\text{ATom})$), to assess model performance. To ensure the robustness and inter-
pretability of these ratios, we apply a thresholding criterion whereby only ATom measurements with positive values (> 0) are



included in the analysis. This step avoids artificially large ratios or undefined logarithmic values that can arise from dividing by zero or by near-zero values — particularly those resulting from background subtraction in the ATom dataset, which may yield small positive or negative values. Prior to further analysis, only the spatial and temporal points with valid ATom measurements are retained, and the corresponding model values at those locations are also preserved to ensure consistency. Data points with

invalid ATom values are excluded from both datasets. While this approach may introduce a slight positive bias in the statistical summaries (e.g. mean and median) of the observational dataset, it ensures that the ratio-based comparisons remain physically meaningful and are not dominated by noise near the detection limit. The variables most affected by this thresholding are those frequently measured near instrument detection limits — especially nitrate aerosol mass throughout the vertical column, as well as ammonium and seasalt aerosol mass above the marine boundary layer (MBL).

The aerosol number size distribution data are further processed to derive the total aerosol number concentration and the size-resolved number concentrations for the four modes: nucleation, Aitken, accumulation, and coarse, to be consistent with the processed ATom data. Similarly, the modelled aerosol chemical composition is processed to facilitate comparison with the AMS measurements of submicron aerosols. The AMS size range is defined by the performance of the aerodynamic lens used as the instrument inlet, and hence, operational transmission is defined by its vacuum aerodynamic diameter, which differs

from the optical/geometric diameter used by aerosol sizers and models (Guo et al., 2021; Brock et al., 2021; Kim et al., 2025). Aerodynamic diameter is typically smaller than geometric diameter, depending on particle density. Moreover, the transmission efficiency of particles larger than approximately 500 nm in aerodynamic diameter decreases log-linearly with size and is not unity. As a result, the AMS measures only a fraction of aerosol mass with a geometric diameter below 1 $\mu$m. To enable direct comparison between model outputs and AMS observations, a real-time transmission correction based on the geometric

diameter is applied following Guo et al. (2021).

For sulfate, organic, nitrate, and ammonium components, their total mass is calculated by summing the mass in the model's nucleation and Aitken modes with the mass fraction of aerosols from the accumulation and coarse modes after applying transmission correction. For seasalt, which is not present in the model's nucleation and Aitken modes, only the mass fraction of aerosols smaller than 1 $\mu$m, and corrected by AMS transmission, from the accumulation and coarse modes is included

in this comparison. Therefore, the total submicron aerosol mass comparison presented in this study is based on the sum of sulfate, organic matter, nitrate, ammonium, and seasalt components only. It should be noted that black carbon and dust are not included in this chemical composition comparison, as aerosol mass spectrometer measurements do not typically quantify these refractory components. However, these species are included in the model simulations and contribute to the evaluated aerosol number size distributions.

Following the recommendation of Williamson et al. (2021), we exclude potential stratospheric air from the model-ATom comparison when either the model or observation indicated O$_3$ > 250 ppbv and RH < 10 %. After this stratospheric filtering, the analysis is conducted across three altitude-based layers: (1) 0 - 2 km (MBL), (2) 2 - 8 km (lower to mid free troposphere), and (3) 8 - 12 km (upper troposphere, UT).

Data during aircraft takeoff and landing are not excluded in the analysis of this study, as in the original dataset provided by

Brock et al. (2019). However, this inclusion is not expected to significantly affect the results. Most quantitative analyses in




this study - such as tables and vertical profiles — use median values, which are robust to outliers. The only exception is the curtain plots, which display mean values due to the limited number of data points in each grid cell, making median estimates statistically unreliable. Importantly, since takeoff and landing events are geographically localised, their influence on the global-scale patterns depicted in the curtain plots is minimal. Therefore, the overall impact on our analysis is expected to be negligible.


## 3 Results

This study evaluates the performance of the model simulations for many aerosol relevant parameters, as listed in Table 3, against the ATom observations. The data presented in Table 3 are median ratios of modelled to observed (ATom) values. For simplicity, these model-to-ATom ratios are referred to as "ratios" throughout the manuscript, unless otherwise specified. The

values presented in this summary table are calculated using all available data points below 2 km altitude, an altitude covering the marine boundary layer (MBL, for simplicity defined as < 2 km in this study) and, depending on location, parts of the lower free troposphere. In the Appendix, we further tabulate the model-to-ATom ratios for the 2 - 8 km and 8 - 12 km altitude ranges (Tables A1 and A2, respectively).

Since a single median value is insufficient to capture the full distribution of the data, our results and discussions will

also feature model-ATom comparisons presented as curtain plots (latitudinal and vertical distributions) of both the original measured/modelled fields and the ratios of modelled to observed values. The analysis is further complemented by vertical profile comparisons for specific parameters, probability density functions of aerosol distribution, and horizontal spatial maps at selected altitudes.

### 3.1 Vapour concentrations


The observed and modelled mixing ratios of key precursor vapours and oxidants (DMS, $SO_2$, $NH_3$, $O_3$, and OH) are presented in Figure 1, while the corresponding ratios of modelled to observed values are shown in Figure 2. All mixing ratio data in these figures are displayed on a logarithmic scale to better visualise the magnitude of differences between the model simulations and observations.


#### 3.1.1 Oxidants

The measured $O_3$ and OH mixing ratios both exhibit a distinct inter-hemispheric asymmetry, with consistently higher values observed in the Northern Hemisphere (NH) compared to the Southern Hemisphere (SH). This asymmetry is well-understood to be driven by higher anthropogenic emissions of ozone precursors, such as nitrogen oxides ($NO_x$) and hydrocarbons, in the

NH (Wang and Jacob, 1998). Regarding latitudinal distribution, observed $O_3$ mixing ratios are generally lowest in the tropics and increase towards higher latitudes, a pattern consistent with recent compilations of oceanic and polar $O_3$ data (Kanaya





**Table 3.** Model-to-ATom median ratios for environmental conditions, precursor vapours, aerosol number size distributions, and chemical composition in the marine boundary layer (0 - 2 km altitude). Ratios are calculated using all available ATom data below 2 km for comparison with different UKESM1.1 configurations implementing various SA (sulfuric acid)-based nucleation schemes (see Table 2 for simulation details). The median values are calculated from the point-by-point model-to-ATom ratios (i.e. the ratio is computed at each location or time point, followed by taking the median), and therefore do not necessarily equal the ratio of the median model value to the median ATom value. Values greater than 1 indicate model overestimation, while values less than 1 indicate model underestimation relative to ATom observations. Environmental conditions include T, RH, and CS of dry aerosols. Precursor vapours include DMS, $SO_2$, $NH_3$, $O_3$, and OH. Aerosol number concentrations are reported for nucleation (dry diameter < 10 nm), Aitken (10 - 100 nm), accumulation (100 - 1000 nm), and coarse (1000 - 10,000 nm) modes, along with total number concentration. Chemical composition includes sulfate, organic matter, ammonium, nitrate, and seasalt mass concentrations, with total mass representing their sum. Missing values indicate that ammonium and nitrate components are not included in simulations without the ammonium nitrate scheme (SA-$H_2O$[default] and SA-$NH_3$-noNit). Similar tables for the 2 - 8 km and 8 - 12 km altitude ranges are provided in the Appendix (Tables A1 and A2, respectively).

| | **H$_2$SO$_4$(SA)-based schemes** | | | | | | | | |
| | H$_2$O(default) | NH$_3$-noNit | NH$_3$-slow | NH$_3$-Lana | NH$_3$(benchmark) | NH$_3$-CS2 | NH$_3$-MSA | NH$_3$-IPSOA | NH$_3$-IPSOA×10 |
|---|---|---|---|---|---|---|---|---|---|
| **Temperature** | 0.99 | 0.99 | 0.99 | 0.99 | 0.99 | 0.99 | 0.99 | 0.99 | 0.99 |
| **RH** | 1.08 | 1.08 | 1.08 | 1.08 | 1.08 | 1.08 | 1.08 | 1.08 | 1.08 |
| **CS dry** | 1.12 | 1.24 | 1.27 | 1.26 | 1.24 | 1.23 | 1.27 | 1.23 | 1.28 |
| **DMS** | 2.35 | 2.35 | 2.48 | 3.45 | 2.59 | 2.61 | 2.59 | 2.52 | 2.62 |
| **SO$_2$** | 1.96 | 1.90 | 1.98 | 2.13 | 2.00 | 0.91 | 1.89 | 1.99 | 1.96 |
| **NH$_3$** | 371.90 | 378.24 | 11.28 | 6.40 | 12.25 | 16.53 | 7.42 | 11.51 | 11.15 |
| **O$_3$** | 1.05 | 1.05 | 0.99 | 0.99 | 0.99 | 0.98 | 0.98 | 0.99 | 0.98 |
| **OH** | 1.39 | 1.38 | 1.31 | 1.31 | 1.31 | 1.32 | 1.30 | 1.29 | 1.29 |
| **Nucleation** | 0.09 | 0.41 | 0.11 | 0.08 | 0.10 | 0.08 | 0.04 | 0.09 | 0.09 |
| **Aitken** | 0.57 | 0.99 | 0.54 | 0.44 | 0.43 | 0.54 | 0.41 | 0.42 | 0.41 |
| **Accumulation** | 1.10 | 1.25 | 1.28 | 1.26 | 1.20 | 1.23 | 1.24 | 1.21 | 1.21 |
| **Coarse** | 0.67 | 0.66 | 0.69 | 0.69 | 0.70 | 0.70 | 0.70 | 0.70 | 0.72 |
| **Total number** | 0.58 | 0.93 | 0.62 | 0.52 | 0.53 | 0.59 | 0.49 | 0.52 | 0.49 |
| **Sulfate** | 1.08 | 1.12 | 1.04 | 1.03 | 0.98 | 0.92 | 1.08 | 0.99 | 0.93 |
| **Organic** | 1.40 | 1.50 | 1.44 | 1.36 | 1.37 | 1.18 | 1.28 | 1.34 | 1.73 |
| **Ammonium** | – | – | 2.93 | 3.13 | 3.17 | 2.91 | 3.25 | 3.20 | 2.95 |
| **Nitrate** | – | – | 3.86 | 9.34 | 9.61 | 6.48 | 7.23 | 10.09 | 9.43 |
| **Seasalt** | 2.09 | 2.21 | 0.98 | 1.05 | 1.03 | 1.00 | 1.04 | 1.01 | 1.03 |
| **Total mass** | 1.24 | 1.29 | 1.23 | 1.25 | 1.24 | 1.14 | 1.22 | 1.21 | 1.27 |

et al., 2025). In contrast, observed OH concentrations show approximately the opposite latitudinal trend to $O_3$, with the highest





values found in the tropics. This inverse latitudinal distribution is primarily attributed to the stronger solar radiation in the tropics which leads to more rapid photochemical destruction of $O_3$, which in turn drives higher OH production rates.

The model-to-ATom ratios for $O_3$ and OH in Table 3, show relatively little variation between the different model simulations performed in this study. Comparing the benchmark SA-$NH_3$ simulation with the ATom measurements reveals that $O_3$ is generally overestimated by the model in the tropics throughout the vertical profile, transitioning to a slight underestimation in the polar regions. Interestingly, despite the asymmetry in the absolute observed concentrations, the pattern of modelled $O_3$ discrepancies relative to observations appears broadly symmetrical between the hemispheres. Quantitatively, the median over-
estimation of $O_3$ in the tropics (25 °S to 25 °N, full altitude range unless otherwise specified) is 28 % (i.e. model-to-ATom ratio of 1.28), while the median underestimation in the polar regions (60 - 90 °N/S) is 15 % (i.e. model-to-ATom ratio of 0.85). However, the model discrepancies for OH differ significantly from those for $O_3$ and exhibit a strong inter-hemispheric asymmetry. North of approximately 50 °S latitude, modelled OH concentrations are generally overestimated by around 7 %, whereas south of 50 °S, modelled OH is substantially underestimated, by 53 %.

The reason for this systematic underestimation of OH concentrations south of 50 °S by the model is currently unclear and warrants further investigation in future studies. However, some potential contributing factors can be explored using the data presented. The primary photochemical production pathway for OH involves the photolysis of $O_3$ followed by the reaction of the resulting $O(^1D)$ atom with water vapour ($H_2O$). Our evaluation of relative humidity (Figure 3) indicates a general model overestimation south of 50 °S. Given that modelled $O_3$ is only slightly underestimated in this region (by 19 %), inaccuracies
in the modelled concentrations of the primary precursors ($O_3$ and $H_2O$) seem unlikely to be the main drivers of the substantial OH underestimation. However, uncertainties in the modelled photolysis rate of $O_3$ ($J(O^1D)$) represent one factor to be examined, as this directly impacts the OH production rate. Additionally, the primary loss processes for OH involve reactions with methane ($CH_4$) and carbon monoxide (CO); therefore, the accuracy of their simulated concentrations is also important. Other factors potentially contributing to the discrepancy include inaccuracies in modelled $NO_x$ concentrations (which influence OH
recycling) or other unexpected OH loss processes in the model.

### 3.1.2   Sulfur species

As key precursors for atmospheric $H_2SO_4$, DMS and $SO_2$ are also evaluated against ATom observations in this study (Figure 1). The measured DMS mixing ratios during ATom are generally highest in the MBL, with concentrations decreasing with
increasing altitude. Within the MBL, measured DMS mixing ratios are relatively higher in the tropics compared to mid-latitude and polar regions, and the distribution appears essentially symmetrical between the hemispheres. Median DMS mixing ratios measured by ATom were around 5.6 pptv below 2 km altitude and 0.6 pptv above 2 km. In contrast to DMS, observed $SO_2$ mixing ratios exhibit a more homogeneous distribution throughout the marine atmosphere, with an overall median value of 11.9 pptv. However, it must be noted that the $SO_2$ data presented here are solely from the ATom4 campaign (Spring 2018) and
thus may not fully represent the complete annual cycle of $SO_2$ distribution.





Comparing modelled DMS with observations reveals that simulations utilising the interactive DMS emission scheme consistently overestimate observed mixing ratios by roughly a factor of 19.80 south of 40 °S across all altitudes (SA-NH$_3$ simulation). The SA-NH$_3$-Lana simulation, which employs a climatological DMS emission scheme, shows marginally better agreement in this high southern latitude region, although it still overestimates observed DMS by a factor of 11.15. Conversely, considering

the global MBL, the median modelled overestimation factor for DMS is lower in the interactive scheme simulation (SA-NH$_3$, factor of 2.59) compared to the climatology simulation (SA-NH$_3$-Lana, factor of 3.45) in Table 3. Therefore, judged by median performance, the interactive DMS emission scheme appears to perform better within the MBL globally but significantly worsens the model performance at high southern latitudes compared to the climatology. It is worth noting, however, that the spatial patterns of the model-ATom discrepancies for DMS are remarkably similar between simulations using the interactive

and climatological emission schemes (Figure 2). This similarity suggests that resolving uncertainties in DMS emissions alone is insufficient to correct the model biases in atmospheric DMS distribution. Accurate representation of atmospheric chemical transformation and loss processes is likely of equal, or even higher, importance. Similarly, modelled SO$_2$ mixing ratios south of 40 °S (full altitude range) are also overestimated relative to ATom4 data, by 32 % in the SA-NH$_3$ simulation and by a lesser amount, around 21 %, in the SA-NH$_3$-Lana simulation.

While switching the DMS emission scheme primarily impacts modelled DMS and SO$_2$ concentrations in a latitude-dependent manner (particularly at high southern latitudes), changing the core tropospheric chemistry scheme from Strat-Trop to CS2 exerts a global influence on simulated SO$_2$ mixing ratios. Simulated SO$_2$ mixing ratios are consistently lower in the SA-NH$_3$-CS2 simulation compared to simulations employing the Strat-Trop scheme (Figure 2 and Tables 3,A1,A2). Consequently, the SO$_2$ simulation in SA-NH$_3$-CS2 shows reasonable agreement with observations (slight overestimation) between

approximately 40 °S and 60 °N, but tends towards underestimation outside this latitude range. This suggests that the modified DMS oxidation chemistry within the CS2 scheme significantly alters the simulated SO$_2$ budget compared to Strat-Trop (Table 1). However, it must be pointed out that this apparent improvement in the SO$_2$ simulation with CS2 is not necessarily indicative of a better overall representation of sulfur chemistry, given that DMS remains significantly overestimated by the model in most regions. Therefore, the seemingly better SO$_2$ agreement might arise from compensating errors within the model's sulfur cycle

representation, an issue warranting further investigation.

### 3.1.3 Ammonia

As previously reported by Nault et al. (2021), atmospheric NH$_3$ mixing ratios estimated during ATom are generally low, often below 1 pptv (Figure 1). The highest NH$_3$ mixing ratios were observed in the tropical MBL (25 °S to 25 °N; altitude < 2 km)

with a median value of 6.6 pptv. Outside this tropical MBL region, observed NH$_3$ mixing ratios frequently dropped below 1 pptv, at which NH$_3$ would typically have a negligible impact on aerosol nucleation processes involving H$_2$SO$_4$.

In contrast to observations, modelled NH$_3$ mixing ratios in the default SA-H$_2$O simulation are globally overestimated by several orders of magnitude compared to observations (Figure 2). For example, the median NH$_3$ mixing ratio in the tropical MBL is simulated to be 215.4 pptv in this configuration, 32.64 times higher than the observed median. In the SA-NH$_3$-noNit





simulation (which includes $H_2SO_4 - NH_3$ aerosol nucleation but not the ammonium nitrate scheme), the median tropical MBL $NH_3$ mixing ratio remains high at 219.0 pptv. However, after implementing the ammonium nitrate scheme developed by Jones et al. (2021) (which enables the uptake of $NH_3$ by acidic aerosols), the modelled median $NH_3$ mixing ratios in the tropical MBL are significantly reduced, to 26.3 pptv and 26.8 pptv in the SA-$NH_3$-slow and SA-$NH_3$ benchmark simulations, respectively. These values are considerably closer to the observed median. This result strongly suggests that the uptake of $NH_3$ by acidic

aerosols to form ammonium is a significant sink for gaseous $NH_3$ in the remote marine atmosphere, and that simulated $NH_3$ concentrations are highly sensitive to the representation of this process. It should be noted, however, that even in the SA-$NH_3$ benchmark simulation (incorporating the ammonium nitrate scheme), global atmospheric $NH_3$ is still overestimated by a factor of 12.25 when compared to ATom data in the MBL (Table 3). Using the slow nitrate uptake coefficient, the SA-$NH_3$-slow scheme does not improve the ammonia simulation, with the model-to-ATom $NH_3$ ratio remaining at 11.28. A similar trend

of atmospheric $NH_3$ overestimation is consistently observed in other chemical transport models as well (Nault et al., 2021). This persistent, large overestimation of gaseous $NH_3$ inevitably drives an even larger overestimation in the formation rate of new aerosol particles via the $H_2SO_4 - NH_3$ nucleation pathway, as will be discussed later. Therefore, further constraining modelled $NH_3$ mixing ratios, likely through improved emission inventories or more detailed representation of its uptake and loss processes, remains a critical area for model development to improve aerosol simulations of $NH_3$ in the remote marine

atmosphere (Ge et al., 2021).





**Figure 1.** Curtain plots of precursor vapour and oxidant concentrations along ATom flight tracks. Mean concentrations of DMS, $SO_2$, $NH_3$, $O_3$, and OH are shown for ATom observations (first row) and selected model simulations (subsequent rows; see Table 2 for simulation details). Model outputs are interpolated to ATom flight coordinates and times. All data are displayed on a logarithmic scale to enhance visualisation of concentration variations spanning multiple orders of magnitude. Colour scales are consistent within each column to facilitate direct comparison between observations and model simulations.





**Figure 2.** Curtain plots of model-to-ATom ratios for precursor vapour and oxidant concentrations along ATom flight tracks. Mean ratios of modelled to observed concentrations for DMS, $SO_2$, $NH_3$, $O_3$, and OH are shown for selected model simulations (see Table 2 for simulation details). Model outputs are interpolated to ATom flight coordinates and times. Data are displayed on a logarithmic scale with a diverging colour scheme where values greater than 1 indicate model overestimation and values less than 1 indicate model underestimation. Colour scales are consistent within each column to facilitate comparison between different model configurations.

## 3.2 Environmental conditions

Besides the concentrations of precursor vapours and oxidants, environmental conditions such as T and RH are also crucial for airborne aerosol formation processes. Generally, lower temperatures and higher relative humidities are more favourable for new particle formation, thus promoting aerosol nucleation processes (Kirkby et al., 2011; Dunne et al., 2016).

Since the model simulations are nudged towards reanalysis meteorological fields, the simulated temperatures are generally in good agreement with ATom observations. For example, the median model underestimation of temperature in the SA-$NH_3$





simulation is only 0.8 K globally. This relatively small temperature bias is not expected to significantly impact calculated aerosol nucleation rates, as their temperature dependence is generally modest for temperature changes of this magnitude. Both

simulated temperature and RH show negligible differences between the various model configurations tested in this study, as the implemented changes in aerosol formation schemes do not significantly affect these meteorological variables on the timescales considered. However, while temperature is generally well represented, modelled RH is consistently overestimated by UKESM1.1 across most of the sampled atmosphere (Figure 4). For example, the median RH overestimation is 1.08 (model-to-ATom ratio) in the MBL, increasing to 1.13 - 1.15 in the UT (Table A2). The overestimation of RH in the UT will inevitably

lead to an overestimation of aerosol nucleation rates from $H_2SO_4$-$NH_3$ based pathways, given the RH-dependence of these processes (Dunne et al., 2016).

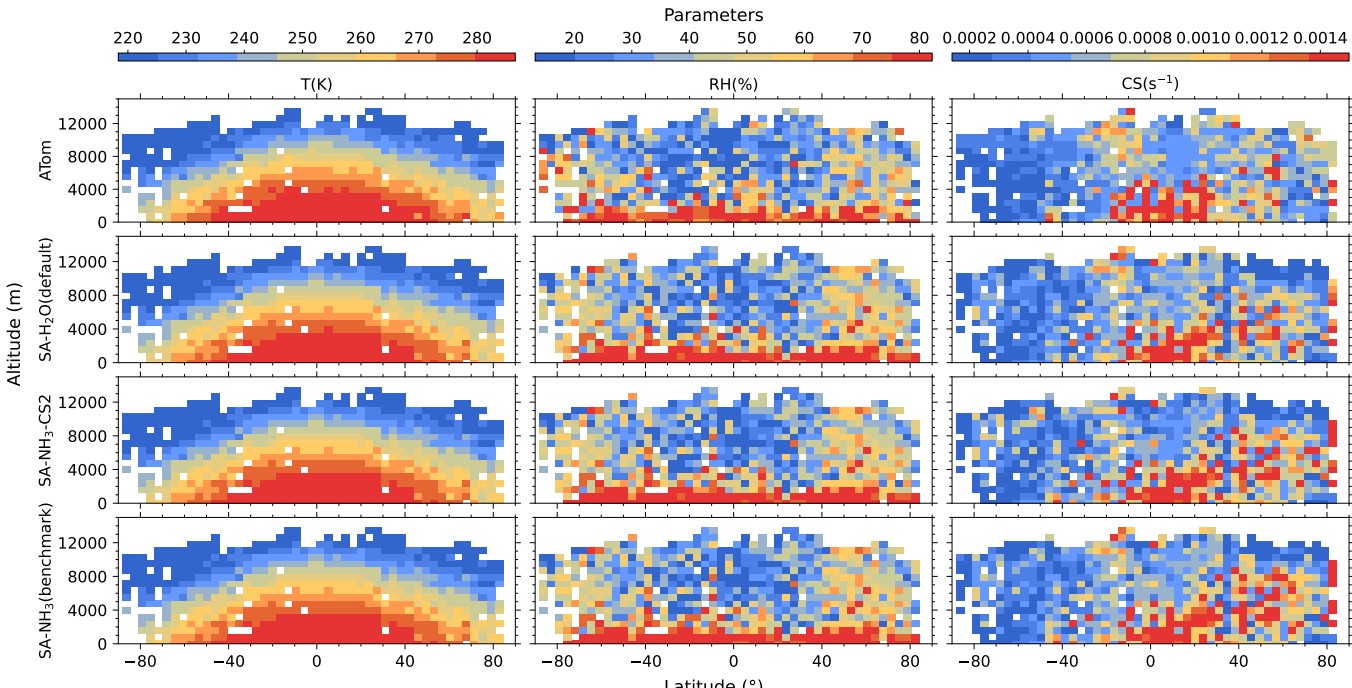

**Figure 3.** Curtain plots of environmental conditions along ATom flight tracks. Mean T, RH, and condensation sink (CS) of dry particles are shown for ATom observations (first row) and selected model simulations (subsequent rows; see Table 2 for simulation details). Model outputs are interpolated to ATom flight coordinates and times. All data are displayed on linear scale. Colour scales are consistent within each column to facilitate direct comparison between observations and model simulations.




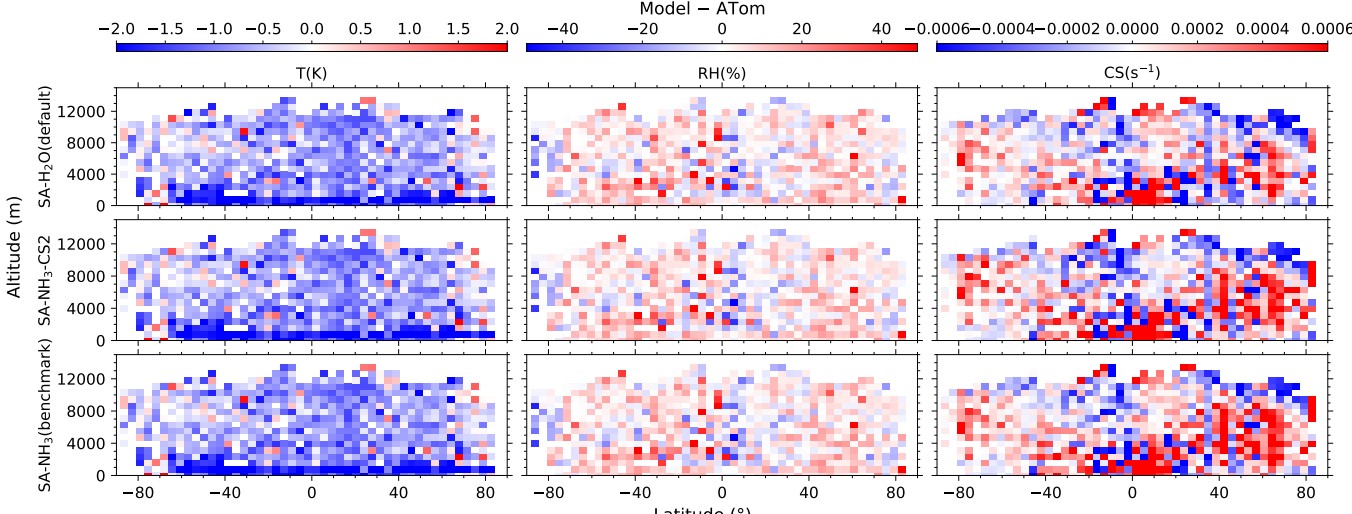

**Figure 4.** Curtain plots of model-to-ATom differences (model - ATom) for environmental conditions along ATom flight tracks. Mean differences of modelled to observed values for T, RH, and CS are shown for selected model simulations (see Table 2 for simulation details). Model outputs are interpolated to ATom flight coordinates and times. Data are displayed on linear scale with a diverging colour scheme. Colour scales are consistent within each column to facilitate comparison between different model configurations.

## 3.3 Aerosol number size distribution

The observed aerosol number concentrations within each aerosol mode from the ATom measurements, alongside those from selected model simulations, are presented in Figure 5. Ratios of modelled to observed values are similarly shown in Figure 6. While Ranjithkumar et al. (2021) reported a comparison of modelled total aerosol number concentration vertical profiles with ATom observations, this study focuses on a mode-resolved comparison across all four modes (nucleation, Aitken, accumulation, and coarse), as this modal separation is necessary to understand their distinct sources, transformation pathways, and climate impacts.

### 3.3.1 Nucleation and Aitken mode aerosols

The ATom measurements reveal a ubiquitous presence of nucleation mode aerosols throughout the marine atmosphere (Brock et al., 2019; Williamson et al., 2019). The concentrations of nucleation mode aerosols are largely symmetrical between the hemispheres. Another notable feature of the observed nucleation mode aerosol distribution is that its vertical variation is considerably larger in the tropics than in mid-latitudes and polar regions (Figure 7). For example, nucleation mode aerosol concentrations measured at approximately 12 km altitude in the tropics are around two orders of magnitude higher than those at the marine surface level, whereas this vertical gradient in mid-latitudes and polar regions is typically less than one order of magnitude. This phenomenon has previously been associated with strong NPF events occurring in the tropical UT, as





proposed by Clarke et al. (2013); Williamson et al. (2019). Similar to the nucleation mode, Aitken mode aerosols also exhibit a
comparable vertical trend in the tropics, with enhanced concentrations observed in the UT relative to the lower troposphere. One
notable difference between the nucleation and Aitken mode distributions is that Aitken mode aerosols show consistently lower
concentrations within the MBL across all latitudes. This reduction in boundary layer Aitken mode aerosols is likely explained
by enhanced coagulation of nucleation mode aerosols with larger accumulation and coarse mode aerosols. This interpretation
is supported by the enhanced accumulation and coarse mode aerosol concentrations observed in the MBL (Figure 5), which
are significantly higher than those in the UT, providing a substantial coagulation sink.

Comparing the default simulation (SA-$H_2O$) with ATom observations (Figures 6 and 7), the bias in nucleation mode aerosol
concentrations shows a clear dipole pattern: a significant overestimation globally above approximately 4 km, and a strong
underestimation below this altitude. For example, the default simulation overestimates nucleation mode aerosols by roughly
one order of magnitude at 12 km altitude (Figure 7), while it underestimates them with a ratio of 0.09 in the MBL (Table 3).
The overestimation of the nucleation mode in the UT by the default simulation is attributed to several factors. The primary
factor is that the theoretical binary $H_2SO_4 - H_2O$ nucleation mechanism from Vehkamäki et al. (2002) itself overestimates the
neutral $H_2SO_4 - H_2O$ nucleation rate by up to three orders of magnitude in the UT, as highlighted by Yu et al. (2020). Other
secondary factors include the model's overestimation of $SO_2$ and RH (discussed in previous sections), all of which contribute
to an increased nucleation rate. Taking the tropical UT (8 - 12 km, 25 °S to 25 °N) as an example, the model-to-ATom ratios
of $SO_2$ and RH are 1.46 and 1.12, respectively, in the default simulation. The only evaluated factor that might partly counter
this tendency to overestimation is the condensation sink, which the model overestimates by 11 % in the tropical UT; however,
this effect is outweighed by the other contributing factors.

Implementing the $H_2SO_4 - NH_3$ nucleation scheme without the ammonium nitrate scheme (SA-$NH_3$-noNit simulation) does
not improve model skill in simulating aerosol number size distribution in the tropical UT; the overestimation of nucleation
mode aerosols remains high, similar to that in the default SA-$H_2O$ simulation (Figure 6). On the other hand, the SA-$NH_3$-
noNit simulation shows nucleation mode aerosol concentrations more than 4 times higher in the MBL compared to the default
simulation, due to the aerosol nucleation enhancement from $H_2SO_4 - NH_3$ nucleation mechanism (Table 3). However, despite
this enhancement, the simulated nucleation mode aerosol concentrations are still 2.41 times lower than the ATom observations
in the MBL. Furthermore, it must be noted that this apparent improvement in predicting MBL nucleation mode aerosols in
the SA-$NH_3$-noNit simulation is concurrent with a enormous overprediction of $NH_3$ by the model, which is higher than ATom
observations by a factor of 378.24 (Table 3). After implementing the ammonium nitrate scheme, with either a slow (SA-$NH_3$-
slow) or fast (SA-$NH_3$ benchmark) nitric acid uptake coefficient, the overestimation of $NH_3$ is reduced to factors of 11.28
and 12.25, respectively. However, the underestimation of nucleation mode aerosols worsens, with ratios of 0.11 and 0.10 for
these simulations, respectively. This represents a marginal enhancement compared with the default $H_2SO_4 - H_2O$ nucleation
mechanism over pristine marine environments.

As expected, the introduction of additional condensable material for growth, such as MSA (SA-$NH_3$-MSA) and isoprene-
derived SOA (SA-$NH_3$-IPSOA and SA-$NH_3$-IPSOA×10), does not significantly improve the nucleation mode aerosol number
bias - these primarily contribute to increasing aerosol mass, rather than number concentrations (Table 3).



In general, the biases in modelled Aitken mode aerosol concentrations show similar patterns to those for the nucleation mode. In the MBL, Aitken mode aerosols are underestimated by a factor of approximately two for all model simulations except the SA-NH$_3$-noNit simulation (Table 3). The Aitken mode aerosol concentration is well simulated in the SA-NH$_3$-noNit simulation. However, as discussed previously, this improved agreement is associated with the significant overestimation of NH$_3$ in this particular model configuration. Despite this major caveat, the SA-NH$_3$-noNit simulation is the only simulation that substantially improves the model skill in predicting both nucleation mode and Aitken mode aerosol concentrations in the MBL, with model-to-ATom ratios of 0.41 and 0.99, respectively. The aerosol probability density function in the MBL for this simulation also shows much closer agreement with the ATom observations than other simulations (Figure 8). This result — where better aerosol predictions require unrealistically high precursor concentrations — indicates that the H$_2$SO$_4$–NH$_3$ mechanism cannot adequately explain observed MBL airborne aerosol formation. Additional or alternative aerosol formation processes are necessary to explain the high nucleation mode aerosol concentrations observed in the MBL.





**Figure 5.** Curtain plots of aerosol number size distribution along ATom flight tracks. Mean nucleation mode (< 10 nm), Aitken mode (10 - 100 nm), accumulation mode (100 - 1000 nm), and coarse mode (1000 - 10,000 nm) aerosol number concentrations are shown for ATom observations (first row) and selected model simulations (subsequent rows; see Table 2 for simulation details). Model outputs are interpolated to ATom flight coordinates and times. All data are displayed on a logarithmic axis to enhance visualisation of the variations in parameters. Colour scales are consistent within each column to facilitate direct comparison between observations and model simulations.





**Figure 6.** Curtain plots of model-to-ATom ratios for aerosol number size distribution along ATom flight tracks. Mean ratios of modelled to observed values for nucleation mode (< 10 nm), Aitken mode (10 - 100 nm), accumulation mode (100 - 1000 nm), and coarse mode (1000 - 10,000 nm) aerosol number concentrations are shown for selected model simulations (see Table 2 for simulation details). Model outputs are interpolated to ATom flight coordinates and times. Data are displayed on a logarithmic scale with a diverging colour scheme where values greater than 1 indicate model overestimation and values less than 1 indicate model underestimation. Colour scales are consistent within each column to facilitate comparison between different model configurations.





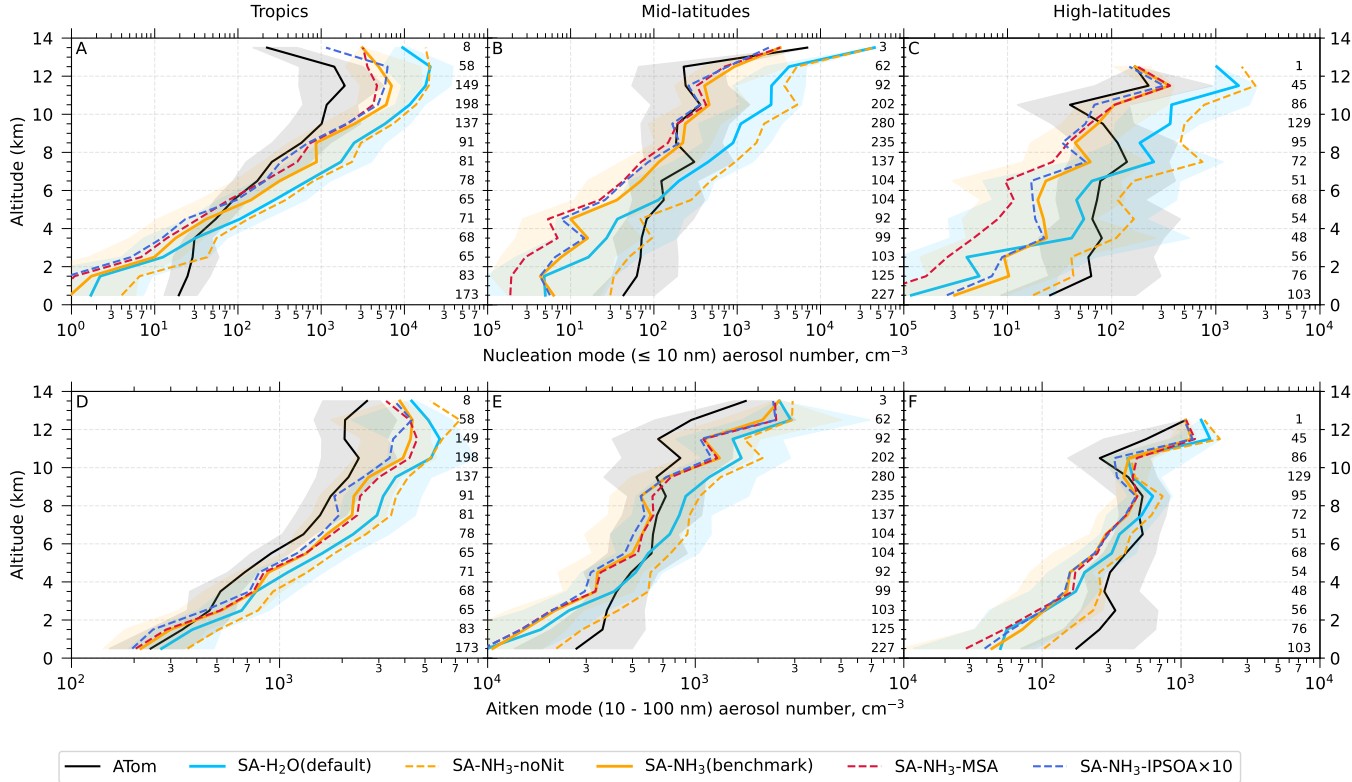

**Figure 7.** Vertical profiles of nucleation mode (< 10 nm) and Aitken mode (10 - 100 nm) aerosol number concentrations (cm$^{-3}$). ATom observations (black lines) and selected model simulations (coloured lines; see Table 2 for simulation details) are presented. Median values are shown for nucleation mode aerosols in the tropics (A, 25 °S to 25 °N), mid-latitudes (B, 25 - 60 °N/S), and high-latitudes (C, 60 - 90 °N/S), and for Aitken mode aerosols in the tropics (D), mid-latitudes (E), and high-latitudes (F). Shaded areas for solid lines represent the interquartile range (25th to 75th percentiles) for both ATom observations and model simulations, indicating the variability in aerosol concentrations. Numbers on the right edge of each panel show the sample size used to calculate median values for each altitude bin at 5-minute temporal resolution.





**Figure 8.** Aerosol number size distribution probability density functions in the marine boundary layer (0 - 2 km altitude) showing median values for ATom observations and selected model simulations. Probability density functions of aerosol number concentration ($dN/d\log D_p$) are shown for ATom observations (gray dots) and model simulations (coloured lines; see Table 2 for simulation details). Gray shade represents the interquartile range of the ATom observations. The x-axis represents aerosol diameter in micrometers, while the y-axis shows the probability density function of aerosol number concentration.





### 3.3.2 Accumulation and coarse mode aerosols

The observed global distributions of accumulation mode and coarse mode aerosols differ significantly from those of the nucleation and Aitken modes. Generally, regions with high Aitken mode aerosol concentrations are typically associated with low concentrations of accumulation mode aerosols, suggesting a somewhat inverse relationship in their spatial distributions (Figures 5 and 6). This relationship is partly regulated by the amount of condensable vapours available for growth, through which Aitken mode aerosols grow into the accumulation mode. Coarse mode aerosols exhibit significantly higher concentrations in the lower troposphere (below approximately 4 km; Figure 9). This is attributed to their larger size, which limits their atmospheric mobility and makes them more susceptible to deposition processes; thus they are concentrated nearer to their emission sources. Overall, both accumulation and coarse mode aerosol concentrations are significantly higher in the mid- to high-latitudes of the NH compared to the SH, a difference primarily attributed to greater anthropogenic activities in the NH. Emissions of primary aerosols, as well as anthropogenic precursor vapours (such as $SO_2$) and oxidants (such as $O_3$ and OH), are all substantially higher in the NH than in the SH.

Accumulation mode aerosols are notably the only mode consistently overestimated by the presented model simulations in the MBL, with a model-to-ATom ratio of 1.20 in the SA-$NH_3$ benchmark simulation (Table 3). There are several potential reasons for this overestimation: 1) excessive aerosol nucleation and subsequent growth processes in the MBL, 2) excessive primary emissions of accumulation mode aerosols, or 3) inefficient aerosol removal processes. The first reason is unlikely, as both nucleation mode and Aitken mode aerosols are significantly underestimated in the MBL, as discussed previously. The second reason is likely, as the model appears to overestimate primary emissions contributing to accumulation mode aerosols, such as seasalt, in the MBL (supported by aerosol composition data discussed in the next section; see Table 3). The third reason cannot be ruled out, but it cannot be assessed with the data available in this study.

While accumulation mode aerosols are overestimated in the MBL, they are underestimated in the UT, where the availability of condensable vapours strongly limits particle growth (Figure 9). Since $H_2SO_4$ is likely already overestimated in the tropical UT by the model (due to overestimation of $SO_2$ and RH), the condensable vapours driving this additional growth are likely to be oxidised organic compounds, such as isoprene-derived SOA, transported from rainforests to the marine UT, as suggested by Shen et al. (2024) and Curtius et al. (2024). The contribution of organics is further supported by aerosol composition analysis in the next section. The addition of isoprene SOA does reduce the underestimation of accumulation mode aerosols in the tropical UT. For example, the SA-$NH_3$-IPSOA×10 simulation leads to a significant increase in modelled accumulation mode aerosol concentrations in this region, resulting in a model-to-ATom ratio of 0.65. This represents an increase of 0.16 in the ratio compared to the benchmark simulation (which has a ratio of 0.49). Simultaneously, the overestimation of Aitken mode aerosols in the tropical UT is also reduced by these additional condensable vapours, with the SA-$NH_3$-IPSOA×10 simulation yielding a model-to-ATom ratio of 1.33. This is a decrease of 0.24 in the ratio from the benchmark simulation (which has a ratio of 1.57).

Coarse mode aerosols are generally underestimated by the model in the MBL, with a median model-to-ATom ratio of 0.70 in the SA-$NH_3$ benchmark simulation (Table 3). In regions other than the MBL, coarse mode aerosols are generally overestimated



by the model (Figure 6). The most severe overestimation occurs south of 40 °S above the MBL, where the model overestimation
can exceed a factor of 10, although the absolute number concentrations in this region are very low (less than 0.1 cm$^{-3}$). Since
aerosol mass spectrometer measurements of seasalt used in this study do not include aerosols larger than 1 $\mu$m and dust aerosols
are not measured by the AMS we use for model evaluation in this study. It is, however, intriguing to note that this specific region
of coarse mode overestimation coincides with the region where OH is significantly underestimated, while DMS and SO$_2$ are
overestimated by the model (Figure 2). Finding the physical and chemical reasons for this connection is therefore an interesting
avenue for future research.

The combined surface area of accumulation and coarse mode aerosols, and to a lesser extent Aitken mode aerosols, deter-
mines the available surface for vapour condensation, and thus defines the condensation sink (CS). The CS is calculated here
following the methodology of Pirjola and Kulmala (1998), based on the dry aerosol size distribution. This choice is made
because calculating CS based on wet size distributions introduces additional uncertainties related to aerosol composition and
hygroscopicity, and also because aerosol composition is evaluated independently in the subsequent section. The CS typically
exhibits a negative correlation with the aerosol nucleation rate, as a higher CS indicates a larger aerosol surface area for vapour
condensation, thereby reducing the vapour concentration available for aerosol nucleation. In a large fraction of atmosphere
sampled by ATom, the CS is overestimated by the model (Figure 4). In the MBL, the benchmark simulation overestimates CS
by 24 % (Table 3). Considering that the primary production pathway for H$_2$SO$_4$ in the MBL is the reaction of SO$_2$ with OH,
while its major loss process is condensation onto existing aerosols (governed by CS), a simplified steady-state approximation
suggests [H$_2$SO$_4$] $\propto$ ([SO$_2$][OH])/CS. Given that the model overestimation of CS in the MBL is roughly counterbalanced by
the overestimation of OH (by 31 %) in the benchmark simulation (Table 3), the model-to-ATom ratio for the term [OH]/CS is
close to unity. Consequently, the bias in simulated H$_2$SO$_4$ in the MBL is likely dominated by the bias in SO$_2$, suggesting that
H$_2$SO$_4$ is probably overestimated by a factor of 2 (Table 3).



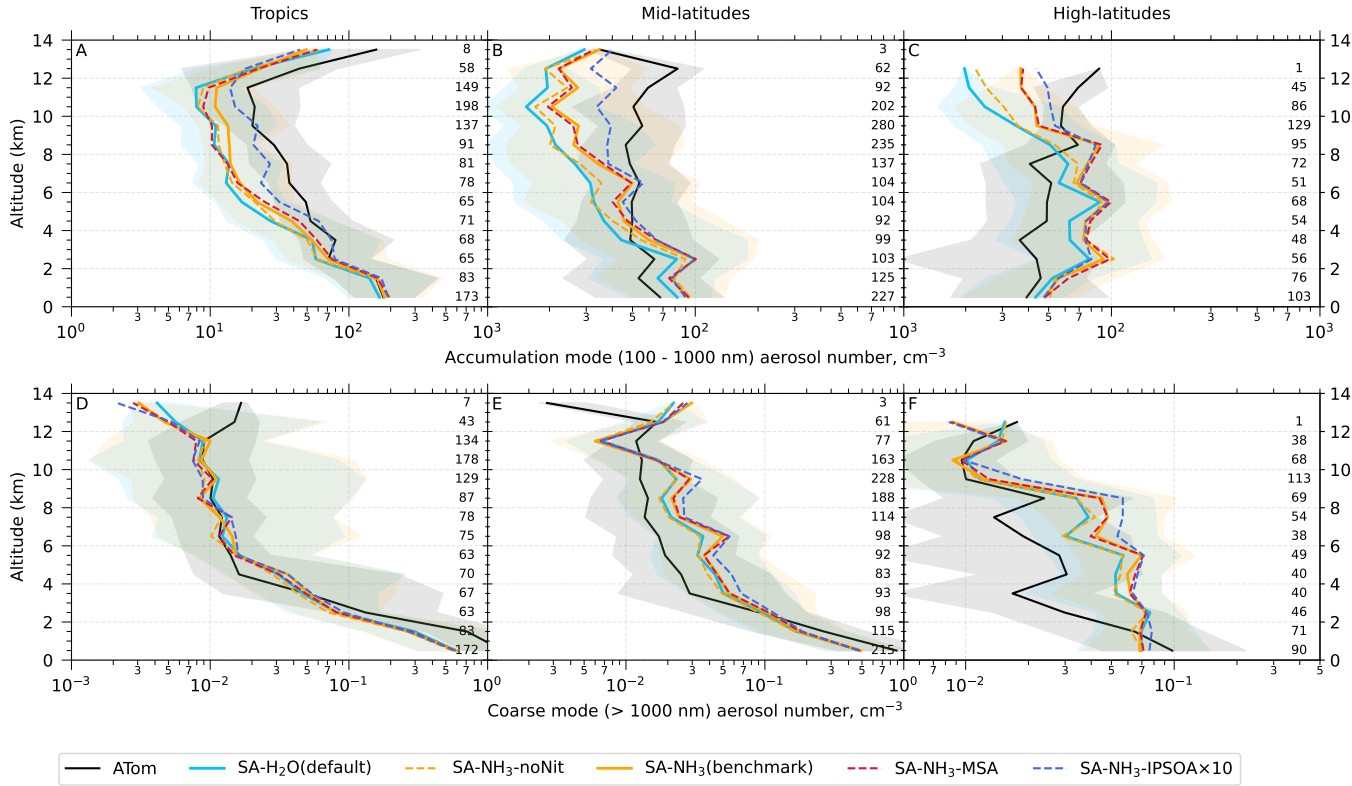

**Figure 9.** Vertical profiles of accumulation mode (100 - 1,000 nm) and coarse mode (1,000 - 10,000) aerosol number concentrations. ATom observations (black lines) and selected model simulations (coloured lines; see Table 2 for simulation details) are presented. Median values are shown for accumulation mode aerosols in the tropics (A, 25 °S to 25 °N), mid-latitudes (B, 25 - 60 °N/S), and high-latitudes (C, 60 - 90 °N/S), and for coarse mode aerosols in the tropics (D), mid-latitudes (E), and high-latitudes (F). Shaded areas for solid lines represent the interquartile range (25th to 75th percentiles) for both ATom observations and model simulations, indicating the variability in aerosol concentrations. Numbers on the right edge of each panel show the sample size used to calculate median values for each altitude bin at 5-minute temporal resolution.

## 3.4 Aerosol composition

In general, model simulations tend to overestimate the MBL submicron aerosol mass (Table 3). This overestimation of aerosol mass occurs despite the general underestimation of total aerosol number concentrations by the model in the MBL. In the benchmark SA-NH$_3$ simulation, for example, the modelled median MBL mass concentrations for sulfate, organic matter, ammonium, and seasalt are 0.198, 0.118, 0.067, and 0.059 $\mu$g m$^{-3}$, respectively. It should be noted that across much of the MBL, nitrate mass concentrations are extremely low, with the AMS frequently reporting zero values. As a result, the spatial coverage of valid nitrate data is limited, and we therefore refrain from reporting its median value. Given that nitrate constitutes only a very small fraction of the total measured and modelled aerosol mass in the MBL, its underestimation does not offset the





overestimation of other components, leading to an overall overestimation of total submicron aerosol mass by a factor of 1.25

in the MBL by the benchmark simulation (Table 3).

This seemingly contradictory result - overestimated aerosol mass alongside underestimated aerosol number concentrations in the MBL - can be reconciled by considering the modal contributions: the model overestimates aerosol concentrations in the accumulation mode (which dominates aerosol mass), while simultaneously underestimating concentrations in the nucleation and Aitken modes (which dominate aerosol number). It should also be noted that the aerosol size distribution in Figure 8

shows that the model's accumulation mode aerosol distribution is skewed towards larger sizes, which is consistent with the overestimation of aerosol mass in the MBL.

### 3.4.1 Sulfate

Except in very specific regions (e.g. the UT south of 40 °S and MBL), sulfate aerosol mass is consistently overestimated by the model, with the magnitude of this overestimation typically being a few tens of percent across much of the sampled atmosphere

(Figures 10 and 11). For example, between 2 - 8 km, the default (SA-H$_2$O) and benchmark (SA-NH$_3$) simulations overestimate sulfate by 42 % and 32 %, respectively (Table A1). The only model simulations that significantly change sulfate aerosol mass concentrations, between 2 - 8 km, are SA-NH$_3$-Lana, SA-NH$_3$-CS2 and SA-NH$_3$-MSA. By modifying the DMS emission scheme, the SA-NH$_3$-Lana simulation increases the sulfate overestimation by 14 %, resulting in a model-to-observation ratio of 1.46. This increase is likely driven by higher DMS emissions from the adopted climatological dataset, relative to the default

interactive scheme, as discussed earlier. On the other hand, the SA-NH$_3$-CS2 simulation reduces the sulfate overestimation from 32 % (in the benchmark) to 6 %, while the SA-NH$_3$-MSA simulation increases the overestimation to 45 % (Table A1). The reduction in sulfate overestimation in the SA-NH$_3$-CS2 simulation is attributed to the lower SO$_2$ concentrations resulting from the CS2 chemistry scheme (see earlier discussions). The additional 13 % increase in bias in the SA-NH$_3$-MSA simulation is due to the inclusion of condensable MSA vapours, which are treated as part of the sulfate aerosol component in the model

for mass accounting, as MSA is not independently treated in the model.

It is important to mention that the most significant changes in sulfate aerosol mass concentrations, relative to ATom, due to these scheme modifications appear to concentrate in relatively pristine environments, such as the tropical UT and the SH (Figure 11). This supports that these changes are primarily driven by changes to the atmospheric chemistry of DMS (in SA-NH$_3$-CS2) and the inclusion of its oxidation products like MSA as condensable species (in SA-NH$_3$-MSA). Anthropogenic primary

sulfate aerosol emissions and direct emissions of SO$_2$, which are more dominant in the NH, are not significantly affected by these specific scheme changes; therefore, sulfate aerosol mass concentrations in the NH are less impacted. This again highlights the need to improve the representation of DMS sources and its atmospheric chemistry in the model for simulating aerosols in the pristine regions of the atmosphere.

One primary reason for this general global overestimation of sulfate aerosol is the model's overestimation of SO$_2$ concen-

825 trations, a key precursor for sulfate aerosol formation. Sulfate aerosol in the model is formed either from primary sources from anthropogenic emissions, or secondary processes such as from the gas-phase condensation of H$_2$SO$_4$, or, more significantly, through aqueous-phase oxidation of SO$_2$ in cloud droplets by H$_2$O$_2$ and O$_3$ (Mulcahy et al., 2020). Consequently, the accuracy





of simulated sulfate aerosol concentrations is also sensitive to the modelled concentrations of other species involved in these pathways, such as gas-phase OH and aqueous-phase oxidants like $H_2O_2$ and $O_3$. Furthermore, the use of a fixed cloud pH value of 5.0 in the model may also affect the efficiency of $SO_2$ uptake (Mulcahy et al., 2020). Since the model's treatment of these multi-phase processes is relatively simplified, further mechanistic implementation and evaluation of these processes are needed to improve the model's representation of sulfate aerosol.

Another potential contributor to the overestimation of sulfate is the underestimation of its removal processes, leading to an overestimated aerosol lifetime. For instance, Mulcahy et al. (2020) reported a sulfate ($SO_4^{2-}$) lifetime of 5.57 days in UKESM1.0, which is around two days longer than the estimate of 3.7 days by Mann et al. (2010). This value also lies at the upper end of the range reported by AeroCom-I models, which spans from 3 to 5.4 days, suggesting that sulfate removal may be too slow in UKESM. It is worth highlighting that sulfate aerosol remains the dominant component of total submicron aerosol mass in the MBL, accounting for 39 % of the total submicron aerosol mass in the benchmark simulation. Accurately representing its sources, from both natural and anthropogenic emissions, is therefore crucial for improving the model's simulation of MBL aerosol properties and, ultimately, CCN number size distributions.

### 3.4.2 Organic matter

The model-ATom discrepancy in organic aerosol mass exhibits a distinct dipole pattern: underestimation in the upper troposphere (8 - 12 km) and the tropical lower to mid free troposphere (2 - 8 km), and moderate overestimation across the rest of the sampled atmosphere (Figures 11 and A2). In the UT, this underestimation is most likely attributed to a lack of significant organic aerosol sources in the model at these altitudes. A clear missing source of organic aerosol in the model's UT is SOA derived from the oxidation products of isoprene, which are not explicitly included as a significant SOA source in the default model configuration. Recent ambient observations and laboratory studies have shown that isoprene oxidation products can contribute significantly to new particle formation and growth in the tropical UT (Shen et al., 2024; Curtius et al., 2024). Although a comprehensive isoprene new particle formation mechanism cannot be implemented in the model at this stage due to a lack of readily available parameterisations, this study attempts to partially account for this missing contribution of isoprene oxidation products to aerosol growth using the SA-NH$_3$-IPSOA and SA-NH$_3$-IPSOA×10 simulations.

The implementation of the explicit IPSOA scheme significantly increases modelled organic aerosol mass concentrations in the tropical UT. The median measured organic aerosol mass concentration during the ATom campaigns in this region is 0.083 $\mu$g m$^{-3}$. The model-to-ATom ratio in the benchmark SA-NH$_3$ simulation for this region is 0.36, indicating an underestimation by around 3 times. With a 3 % SOA mass yield from isoprene oxidation, the SA-NH$_3$-IPSOA simulation improves the model-to-ATom ratio for organic aerosol mass concentration in the tropical UT to 0.41. A 30 % SOA mass yield (SA-NH$_3$-IPSOA×10 simulation) further improves this comparison, increasing the model-to-ATom ratio to 0.86 (Figure 11).

In the MBL, the model simulations generally overestimate organic aerosol mass concentrations, for example by a factor of 1.37 in the benchmark SA-NH$_3$ simulation. The SA-NH$_3$-IPSOA simulation, which introduces a 3 % isoprene SOA yield, results in a model-to-ATom ratio of 1.34 for MBL organic aerosol. This slight reduction in the MBL organic aerosol overestimation, despite adding an isoprene SOA source, is likely due to the concurrent reduction in the prescribed monoterpene





SOA yield. In the default model, the monoterpene SOA yield is scaled by two (effectively 26 %) to implicitly account for missing isoprene SOA. When the explicit isoprene SOA pathway is activated in the SA-NH$_3$-IPSOA simulation, this scaling is removed, and the monoterpene SOA yield reverts to its base value (13 %), which could result in a lower overall SOA pro-
duction from these combined sources compared to the default scaled approach (Table 2). The SA-NH$_3$-IPSOA×10 simulation, on the other hand, significantly worsens the organic aerosol mass overestimation in the MBL, increasing the model-to-ATom ratio to 1.73. This increased overestimation likely results from an isoprene SOA yield (30 %) that is unrealistically high for the warmer MBL conditions, where higher temperatures generally lead to lower SOA yields. Therefore, accurately quantifying the SOA yield from isoprene oxidation, both in the cold UT and warm MBL, is crucial for improving the model's representation
of organic aerosols.

Another notable feature is the generally stronger underestimation of submicron organic aerosol mass in the NH compared to the SH. This disparity is likely due to the model's omission of secondary organic aerosol contributions from anthropogenic precursors, a limitation that warrants further investigation.

### 3.4.3   Ammonium and nitrate

The ammonium and nitrate aerosol scheme, recently developed by Jones et al. (2021), is not included in the default UKESM1.1 configuration. Therefore, the default (SA-H$_2$O) and SA-NH$_3$-noNit simulations do not include these two aerosol components, while the ammonium nitrate scheme is included in all other simulations discussed in this study. ATom observations indicate that the mass concentrations of both ammonium and (inorganic) nitrate are significantly lower than those of sulfate and organic matter globally. This is particularly true for nitrate mass, which exhibits low concentrations globally (Figure 10).

For the region where the nitrate component is above the detection limit of AMS, model simulations generally show an overestimation of nitrate aerosol mass concentrations compared to ATom observations below 8 km (Tables 3 and A1). For example, the benchmark SA-NH$_3$ simulation reports a model-to-ATom ratio of 9.61 in the MBL , indicating a significant overestimation by the model (Tables 3). Using the slow nitric acid uptake coefficient (SA-NH$_3$-slow simulation) reduces this overestimation in the MBL, with the ratio reduced to 3.86. This discrepancy might suggest that the nitric acid uptake coefficient
is still too fast, and using an even slower value could improve the simulation of nitrate aerosol mass concentrations in this NH. Alternatively, the overprediction may be from overestimated anthropogenic NO$x$ emissions, which would lead to excessive atmospheric nitric acid and, consequently, particulate nitrate. A third possible explanation is that the model underestimates the deposition rates of NO$x$ and its oxidation products. Therefore, the exact cause of the nitrate aerosol overestimation in the NH remains uncertain and requires further investigation.

In contrast to nitrate, ammonium aerosol mass concentrations are ubiquitously overestimated by the model throughout the sampled atmosphere when the ammonium nitrate scheme is active (Figures 11 and A1). This overestimation of ammonium likely translates into an overestimation of aerosol pH (i.e. an underestimation of aerosol acidity) globally, consistent with findings for other chemical transport models reported by Nault et al. (2021). In the MBL, the benchmark SA-NH$_3$ simulation overestimates ammonium aerosol mass concentrations by a factor of 3.17 compared to ATom observations (Table 3). Further-
more, this positive model bias for ammonium is even larger above the MBL. For example, the bias increases to a factor of





7.68 between 2 - 8 km altitude. It is worth noting that both ammonium aerosol and gas-phase $NH_3$ concentrations are significantly overestimated by the model in the MBL (Table 3). This pervasive overestimation of ammonium is unlikely to be solely explained by flaws in the implemented ammonium nitrate scheme itself. Improvements to either or both the $NH_3$ emission inventories and the representation of its atmospheric removal processes are needed to enhance the simulation of both gaseous 900 $NH_3$ and particulate ammonium in the model.

### 3.4.4   Seasalt

Seasalt aerosols are a major component of primary marine CCN and play a crucial role in cloud microphysics and climate. The precise relative contributions of seasalt aerosols (primary) and new particle formation processes to the total CCN population remain uncertain. These two major aerosol sources have distinct feedback mechanisms within the climate system and are 905 projected to exhibit different responses to future environmental changes. It is therefore essential to accurately represent both processes in models to enable robust assessments of future changes in CCN, cloud microphysics, and ultimately, climate.

As expected, the measured seasalt concentrates in the MBL (Murphy et al., 2019). However, the simulations without the ammonium nitrate scheme (SA-$H_2O$ and SA-$NH_3$-noNit) overpredict seasalt aerosol mass concentrations in submicron aerosols within the MBL, with the default simulation predicting concentrations 2.09 times higher than observed (Table 3). This overes-910 timation persists consistently across latitudes, from the tropics to high-latitude regions (Figure 11). Additionally, simulations that include the ammonium nitrate scheme show a reduction in sea salt aerosol mass. This decrease is possibly linked to chemical processing, specifically the displacement of chloride by nitrate through heterogeneous uptake of $HNO_3$ on seasalt particles, leading to the formation of hydrogen chloride (HCl) (Jones et al., 2021). However, given the very low nitrate concentrations in the marine boundary layer, it remains uncertain whether sufficient nitrate is available to drive appreciable chloride 915 displacement. Further investigation is needed to better understand the implications of the ammonium nitrate scheme on seasalt conservation in marine environments.

Constraining seasalt emissions, particularly with geometric diameter smaller than 1 $\mu$m, is crucial for accurately understanding the CCN budget in the MBL, as Aitken and accumulation mode aerosols are the major contributors to CCN concentrations. The default model's overestimation of seasalt aerosols in the MBL will likely lead to an exaggerated contribution from primary 920 seasalt to the CCN budget, potentially masking or underestimating the CCN contribution from airborne aerosol formation processes. The default model's overprediction of seasalt aerosols in the marine atmosphere is also supported by a recent study by Venugopal et al. (2025), which found that the default seasalt emission scheme in UKESM overpredicts the dependence of seasalt emissions on wind speed when validated against aerosol optical depth observations.





**Figure 10.** Curtain plots of aerosol composition along ATom flight tracks. Mean sulfate, organic, ammonium, nitrate and seasalt mass concentrations are shown for ATom observations (first row) and selected model simulations (subsequent rows; see Table 2 for simulation details). Model outputs are interpolated to ATom flight coordinates and times. All data are displayed on a logarithmic to enhance visualisation of the variations in parameters. Colour scales are consistent within each column to facilitate direct comparison between observations and model simulations. The SA-H$_2$O (default) simulation does not include ammonium or nitrate in aerosols, thus the absence of data in the corresponding plots.





**Figure 11.** Curtain plots of model-to-ATom ratios for aerosol composition along ATom flight tracks. Mean ratios of modelled to observed values for sulfate, organic, ammonium, nitrate and seasalt mass concentrations are shown for selected model simulations (see Table 2 for simulation details). Model outputs are interpolated to ATom flight coordinates and times. Data are displayed on a logarithmic scale with a diverging colour scheme where values greater than 1 indicate model overestimation and values less than 1 indicate model underestimation. Colour scales are consistent within each column to facilitate comparison between different model configurations. The SA-H$_2$O (default) simulation does not include ammonium or nitrate in aerosols, thus the absence of data in the corresponding plots.

## 4 Discussion

Previous sections have evaluated the model's performance in simulating individual environmental conditions, precursor vapours, aerosol number size distributions, and aerosol chemical composition against ATom observations. Addressing the identified discrepancies is crucial for improving the representation of aerosol processes in the model, with the ultimate goal of





achieving accurate predictions for the correct physical and chemical reasons. In this section, we synthesise our findings to discuss the implications of our model development endeavours and to outline future directions for improving aerosol simulations.

The discussions focus on two specific regions: the UT, particularly between 8 and 12 km altitude, and the MBL, below 2 km altitude. The selection of these two regions is based on several key considerations. Firstly, they are the regions where the model exhibits its most significant biases for numerous evaluated parameters. Secondly, the UT is a region where ATom observations reveal significant enhancements in aerosol number concentrations, often driven by strong new particle formation (Williamson et al., 2019). Thirdly, the MBL is critical due to the sensitivity of marine low-level clouds, a key component of the Earth's

radiative balance, to the MBL aerosol population that acts as CCN.

## 4.1   Marine boundary layer processes

The systematic overestimation of aerosol nucleation precursors ($SO_2$, OH and $NH_3$) alongside the underestimation of nucleation and Aitken mode particle concentrations in the MBL highlights a fundamental inconsistency between current model representations and the actual processes governing airborne aerosol formation in marine environments. This paradox - where

abundant precursor vapours fail to generate sufficient particle numbers - suggests that $H_2SO_4 - H_2O$ nucleation alone cannot account for observed aerosol formation rates in the marine atmosphere.

    The implementation of the $H_2SO_4 - NH_3$ nucleation mechanism with the ammonium nitrate scheme produces mixed results. While this leads to modest improvements in nucleation mode particle concentrations, it worsens the underestimation of Aitken mode aerosols (Table 3) and generally reduces Aitken mode concentrations across the MBL, particularly in pristine

marine regions (Figure 12). In contrast, accumulation mode aerosol concentrations exhibit widespread increases, especially over populated continental regions where elevated $NH_3$ emissions promote both aerosol mass formation and $H_2SO_4 - NH_3$ nucleation.

    Model sensitivity to different schemes is substantial. Adding condensable vapours such as MSA increases accumulation mode concentrations, particularly over the Southern Ocean where DMS emission is high, while simultaneously reducing Aitken

mode concentrations as aerosols grow more rapidly to the accumulation mode (Figure 13). Switching DMS emission schemes or chemistry schemes (from Strat-Trop to CS2) produces significant regional changes, with polar regions being especially sensitive due to their low background aerosol concentrations.

    Beyond aerosol numbers, the default model overestimates total submicron aerosol mass in the MBL. For example, seasalt is overestimated in the default model, potentially leading to exaggerated primary CCN contributions. The concurrent overes-

timation of both gas-phase $NH_3$ and particulate ammonium in the simulations with the ammonium nitrate scheme suggests fundamental issues with $NH_3$ budget representation.

    These findings indicate that the default model's CCN budgets in the MBL are incorrectly described, with excessive dependence on primary emissions such as seasalt masking deficient airborne aerosol formation processes. This imbalance has important implications for preindustrial baseline estimates, as the relative importance of natural versus anthropogenic aerosol

sources may be fundamentally misrepresented. The results strongly support incorporating additional aerosol formation pathways, likely involving iodine oxoacids, organic vapours, or other marine-specific precursors.





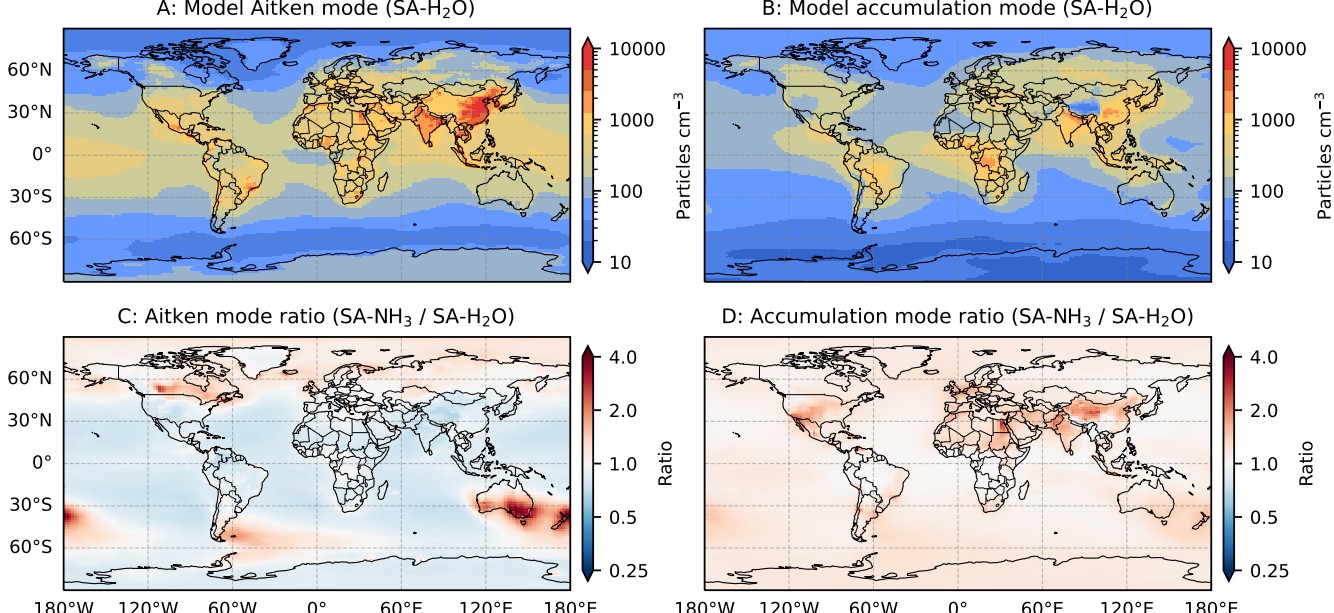

**Figure 12.** Global distribution of marine boundary layer aerosol changes following implementation of the $H_2SO_4 - NH_3$ nucleation mechanism. Panels show monthly mean aerosol number concentrations in the marine boundary layer (0 - 2 km altitude) averaged between July 2016 and June 2018: (A) Aitken mode aerosols from the default SA-$H_2O$ simulation, (B) accumulation mode aerosols from the default SA-$H_2O$ simulation, (C) ratio of Aitken mode concentrations between benchmark SA-$NH_3$ and default SA-$H_2O$ simulations, and (D) ratio of accumulation mode concentrations between benchmark SA-$NH_3$ and default SA-$H_2O$ simulations.




**Figure 13.** Global distribution of marine boundary layer aerosol changes following implementation of the Lana et al. (2011) DMS climatology, MSA condensation and CRI-Strat2 scheme. Panels show monthly mean aerosol number concentrations in the marine boundary layer (0 - 2 km altitude) averaged between July 2016 and June 2018: (A) Aitken mode aerosols from the benchmark SA-NH$_3$ simulation, (B) accumulation mode aerosols from the benchmark SA-NH$_3$ simulation, (C) ratio of Aitken mode concentrations between SA-NH$_3$-Lana and benchmark SA-NH$_3$ simulations, and (D) ratio of accumulation mode concentrations between SA-NH$_3$-Lana and benchmark SA-NH$_3$ simulations, (E) ratio of Aitken mode concentrations between SA-NH$_3$-MSA and benchmark SA-NH$_3$ simulations, and (F) ratio of accumulation mode concentrations between SA-NH$_3$-MSA and benchmark SA-NH$_3$ simulations, (G) ratio of Aitken mode concentrations between SA-NH$_3$-CS2 and benchmark SA-NH$_3$ simulations, and (H) ratio of accumulation mode concentrations between SA-NH$_3$-CS2 and benchmark SA-NH$_3$ simulations.



## 4.2 Upper tropospheric processes

The model's biases for the aerosol number size distribution in the UT are essentially the opposite of those in the MBL. Specifically, the model overestimates nucleation and Aitken mode aerosol concentrations while underestimating accumulation mode aerosol concentrations. This pattern of too many small aerosols and too few large ones may suggest an imbalance in the model between new particle formation and subsequent aerosol growth processes. The UT provides a clearer reflection of the model's performance in simulating these airborne aerosol formation processes, as it is a region where most primary aerosols are not efficiently transported, minimising the confounding influence of primary source biases that are more prominent in the MBL. In the default simulation, the model overestimates nucleation and Aitken mode aerosols by factors of 6.30 and 1.88, respectively, while the model-to-ATom ratio for accumulation mode aerosols is only 0.39 (Table A2). As discussed in earlier sections, this overestimation of small aerosols primarily results from the excessive aerosol nucleation rate predicted by the Vehkamäki et al. (2002) scheme, as well as from contributing factors such as the overestimated concentrations of OH and $H_2O$ (Figures 2 and 4).

Implementing the $H_2SO_4-NH_3$ nucleation mechanism and ammonium nitrate scheme in the benchmark simulation significantly reduces the overestimation of nucleation and Aitken mode aerosols, yielding improved model-to-ATom ratios of 1.79 and 1.39, respectively. Simultaneously, the underestimation of accumulation mode aerosols is also reduced (i.e. improved), with the model-to-ATom ratio increasing to 0.52. This can be understood in terms of mass balance: a lower nucleation rate results in fewer nucleation and Aitken mode aerosols, but the available condensable vapour is then distributed among fewer aerosols, promoting more efficient growth of the remaining population into the accumulation mode. It is worth mentioning that since sulfate aerosol mass is reasonably simulated by the model in the UT (ratio of 1.01 in the benchmark simulation), the missing mass needed to grow aerosols into the accumulation size range and to match observations must be accounted for by other components. As nitrate, ammonium, and seasalt aerosol mass concentrations are all too low in the UT over remote marine atmosphere (Figure 10), organic matter is the only remaining component that can account for this missing mass, consistent with the hypothesis by (Williamson et al., 2019; Kupc et al., 2020).

Indeed, all simulations without the IPSOA scheme show significantly underestimated organic aerosol mass concentrations in the UT, with the benchmark simulation underestimating it with a ratio of 0.44. Isoprene is the most likely candidate to account for this missing organic aerosol mass, as its global emission is large (around 500 Tg yr$^{-1}$), and its oxidation products have recently been discovered to contribute significantly to new particle formation and growth in the UT (Shen et al., 2024; Curtius et al., 2024). Implementing the IPSOA scheme with a high mass yield of 30 % (in the SA-NH$_3$-IPSOA×10 simulation) significantly improves the agreement, increasing the model-to-ATom ratio for organic aerosol mass in the UT to 0.86. The inclusion of the IPSOA scheme results in the largest enhancements in accumulation mode aerosol concentrations above tropical rainforests (Figure 14), a finding consistent with satellite observations (Palmer et al., 2022). It is worth highlighting that the impact of including IPSOA stretches beyond the tropical UT, extending to the mid- and high-latitude UT. This demonstrates that the long lifetime of aerosols in the UT allows them to be transported over great distances, thereby exerting a global impact. Addressing this missing organic aerosol component, and thereby the underestimated accumulation mode aerosol concentrations



in the UT, is crucial for improving the model's representation of ice nucleating particles and cirrus cloud formation, which in turn affects the Earth's longwave radiation balance.

Once again, changing the chemistry scheme from Strat-Trop to CS2 has a large impact on the modelled agreement for Aitken and accumulation mode aerosols in the UT (Figure 14). The reduction in $SO_2$ concentrations in the SA-NH$_3$-CS2 simulation is the clear cause of the reduction in these aerosol populations, as can be seen in Figure 2. This is also supported by the aerosol composition analysis, which shows a corresponding reduction in the sulfate aerosol mass bias, with insignificant changes in other aerosol components (Figure 11).







**Figure 14.** Global distribution of upper tropospheric aerosol changes following implementation of the CRI-Strat2 scheme and isoprene secondary organic aerosol formation scheme. Panels show monthly mean aerosol number concentrations in the upper troposphere (8 - 12 km altitude) averaged between July 2016 and June 2018: (A) Aitken mode aerosols from the benchmark SA-NH$_3$ simulation, (B) accumulation mode aerosols from the benchmark SA-NH$_3$ simulation, (C) ratio of Aitken mode concentrations between SA-NH$_3$-CS2 and benchmark SA-NH$_3$ simulations, and (D) ratio of accumulation mode concentrations between SA-NH$_3$-CS2 and benchmark SA-NH$_3$ simulations, (E) ratio of Aitken mode concentrations between SA-NH$_3$-IPSOA×10 and benchmark SA-NH$_3$ simulations, and (F) ratio of accumulation mode concentrations between SA-NH$_3$-IPSOA×10 and benchmark SA-NH$_3$ simulations.

## 4.3 Recommendations for future model improvements

The evaluation of UKESM against ATom observations in this study has identified several key areas for improving the model's

representation of aerosol microphysics and chemistry in the marine atmosphere. These involve primary aerosol emissions, precursor vapours, nucleation and growth processes, and aerosol chemical composition. While each evaluated parameter has its own specific bias and uncertainties, the two main trends are 1) the overestimated aerosol mass and precursor gasses and





2) the underrepresented aerosol number. This indicates that the airborne aerosol formation rates are underestimated in the UKESM.

Many of the airborne aerosol formation mechanisms are essentially missing in the model, which requires extensive efforts to implement and evaluate these processes. Several candidate mechanisms have been proposed in the literature. The first and most prominent candidate is iodine oxoacids (iodic acid, $HIO_3$, and iodous acid, $HIO_2$), which are widely acknowledged as important for aerosol nucleation in the MBL. Iodine oxoacids are ubiquitously measured in the global atmosphere, from pristine marine to polluted urban environments (Beck et al., 2021; He et al., 2021b, 2023; Zhang et al., 2024). Depending

on the location and season, the concentrations of iodine oxoacids are often comparable to (within one order of magnitude of) or can even exceed those of $H_2SO_4$. Dedicated laboratory experiments have revealed that on a per-molecule basis, iodine oxoacids are significantly more efficient than $H_2SO_4$ at forming new particles (He et al., 2021b). Furthermore, when iodine oxoacids are mixed with $H_2SO_4$ and $NH_3$, the synergistic aerosol nucleation of the sulfur-iodine system is even more efficient than any of the individual components alone (He et al., 2023). A recent global model study has also shown that the inclusion

of iodine oxoacids significantly improved modelled aerosol number concentrations in the MBL (Zhao et al., 2024), although the parameterisation used in that study was mostly a fit to experimental data of He et al. (2021b) and did not include the synergistic effects of the sulfur-iodine system (He et al., 2023). It is worth noting that while Zhao et al. (2024) suggested that iodine oxoacid-driven aerosol nucleation has a negligible impact above the MBL due to rapid scavenging of iodine species in the MBL, ambient observations suggest the widespread presence of iodine compounds throughout the troposphere, including

the UT (Koenig et al., 2020; He et al., 2021b, 2023; Schill et al., 2025). This discrepancy is likely due to the absence of halogen recycling mechanisms in the model used by Zhao et al. (2024).

    The second candidate mechanism is nucleation involving atmospheric amines, which have been shown to have strong synergistic effects with $H_2SO_4$ (Almeida et al., 2013). Sporadic ambient observations of the participation of amines in aerosol nucleation have been reported, for example, near the Antarctic Peninsula (Brean et al., 2021). However, direct measurements of nucleating cluster ions in polar regions typically show a stronger participation of $NH_3$ rather than amines (Beck et al., 2021;

Jokinen et al., 2018), and the global relevance of amines in marine aerosol nucleation is yet to be determined. Global simulations along the ATom flight paths have also suggested a minor contribution of amines to aerosol nucleation in the MBL (Zhao et al., 2024).

    The third candidate mechanism is airborne aerosol formation from organic vapours, such as monoterpenes and isoprene, or

their synergistic nucleation with inorganic acids (e.g. $H_2SO_4$ and iodine oxoacids). Unfortunately, the role of organic vapours in MBL aerosol nucleation remains highly debated, as their marine source strength and subsequent atmospheric concentrations are highly uncertain. For example, Luo and Yu (2010) estimated that global oceanic emissions of $\alpha$-pinene and isoprene using bottom-up estimates are 0.013 TgC yr$^{-1}$ and 0.32 TgC yr$^{-1}$, respectively, significantly lower than oceanic DMS emissions. However, the same study reported that using a top-down approach (forcing the model to match observed organic concentrations)

resulted in much higher global emissions of 29.5 Tg Cyr$^{-1}$ for $\alpha$-pinene and 11.6 TgC yr$^{-1}$ for isoprene, values comparable to global oceanic DMS emissions. More recent in-situ measurements of monoterpenes in the Arctic and Atlantic Oceans suggest concentrations below 5 pptv, supporting an estimated global marine monoterpene emission of 0.16 TgC yr$^{-1}$ (Hackenberg





et al., 2017), which aligns better with the bottom-up estimate of Luo and Yu (2010). Therefore, further constraining marine organic vapour emissions is needed to better represent the role of organic vapours in new particle formation in the MBL. On the other hand, in the UT, the role of isoprene derivatives in new particle formation is unambiguous as reported in Shen et al. (2024); Curtius et al. (2024). The long range transport of the formed aerosols will affect the aerosol populations in the marine atmosphere.

Finally, we outline some of the most important recommendations. While not exhaustive, they provide a solid foundation for future model development efforts.

- Re-evaluate primary aerosol emissions in the MBL with a focus on seasalt and organics.

- Re-evaluate the aerosol deposition schemes in the model.

- Re-evaluate the marine $NH_3$ emission inventory and investigate the bias in the atmospheric $NH_3$ budget both in the gas phase and in aerosols ($NH_4^+$).

- Investigate and improve the representation of OH in the atmosphere, especially in the southern hemisphere.

- Improve the atmospheric DMS chemistry scheme as well as the DMS emissions, particularly in the Southern Ocean, where the model currently overestimates DMS emissions.

- Implement comprehensive halogen chemistry in the model with a focus on representing iodine chemistry and iodine oxoacid production.

- Implement the iodine oxoacid new particle formation mechanism in the model, together with the synergistic effects of sulfur-iodine aerosol nucleation, to improve the representation of new particle formation in the MBL.

- Implement experimentally verified MSA condensation scheme in the model.

- Improve model treatment of organic oxidation pathways, especially in the upper troposphere.

- Implement the isoprene new particle formation mechanism in the model, together with the synergistic aerosol nucleation of isoprene derivatives with $H_2SO_4$ and iodine oxoacids, to improve the representation of new particle formation in the upper troposphere.

## 5 Conclusions

This study provides a detailed evaluation of the United Kingdom Earth System Model version 1.1 (UKESM1.1), using observations from the Atmospheric Tomography (ATom) mission to examine parameters relevant to the aerosol lifecycle in the remote marine atmosphere. By comparing model outputs with measurements of precursor vapours, aerosol number size distributions, and aerosol chemical composition, we identify several key shortcomings in current model's representation of aerosol microphysics and chemistry, and offer targeted recommendations for future development.



Our analysis highlights two contrasting biases in UKESM1.1's treatment of aerosols in the marine boundary layer. The default model overestimates primary submicron seasalt, while underrepresenting airborne aerosol formation processes. At the same time, the model underestimates nucleation and Aitken mode aerosols, with model-to-ATom ratios down to 0.09 and 0.57, respectively. This imbalance suggests that the current cloud condensation nuclei budget in the model is overly reliant on primary emissions, potentially misrepresenting the importance of airborne aerosol formation processes that are likely important in pristine marine regions.

In contrast, the model exhibits an opposite bias in the upper troposphere, generating excessive nucleation and Aitken mode aerosols due to the default $H_2SO_4 - H_2O$ nucleation scheme, but failing to grow them efficiently to accumulation mode sizes relevant for ice cloud formation.

To address these problems, we implement a $H_2SO_4 - NH_3$ nucleation scheme alongside an ammonium nitrate module developed by Jones et al. (2021). This update partly improves performance in simulating aerosol number size distributions in the upper troposphere but does little to resolve the discrepancies in the MBL. The continued underprediction of small aerosols in the MBL - despite substantial overestimation of precursors such as $SO_2$, OH, and $NH_3$ - strongly suggest that the model's representation of airborne aerosol formation processes is incomplete.

Incorporating additional condensable vapours such as isoprene oxidation products, improve the model simulations in the upper troposphere. A modified isoprene secondary organic aerosol scheme with enhanced yields improves observed organic aerosol concentrations in the upper troposphere, consistent with emerging evidence that low temperature isoprene oxidation significantly contributes to aerosol formation and growth aloft (Shen et al., 2024; Curtius et al., 2024). The implemented methanesulfonic acid condensation scheme increases particle mass and alters the particle number size distribution; however, its quantitative impact remains uncertain pending more accurate experimental constraints.

Additionally, the model simulations in this study indicate that the concentrations of cloud condensation nuclei (CCN)-active Aitken and accumulation mode aerosols in the MBL are significantly influenced not only by the implemented aerosol nucleation and growth schemes but also the emission mechanisms for precursor vapours, and the adopted atmospheric chemistry mechanisms. For instance, switching from the interactive DMS emission scheme to the Lana et al. (2011) climatology resulted in substantial changes to both Aitken and accumulation mode aerosol distributions, with reductions south of 30 °S and enhancements in tropical regions. Changing the chemistry scheme from Strat-Trop to CRI-Strat2 leads to significant reductions in $SO_2$ concentrations, which in turn affects aerosol nucleation rates and size distributions throughout the marine atmosphere. These sensitivity tests demonstrate that achieving accurate CCN predictions requires not only mechanistically sound aerosol nucleation and growth parameterisations, but also reliable representations of precursor emissions and their atmospheric transformation pathways. The interconnected nature of these processes means that improvements in one area can be offset by deficiencies in another, highlighting the need for comprehensive, process-based model development rather than isolated parameter adjustments.

These findings carry important implications for climate modeling. The tendency to overestimate primary submicron seasalt in the MBL suggests that the default UKESM1.1 might underestimate the role of anthropogenic aerosol sources from e.g. $SO_2$ in shaping present-day CCN populations. Furthermore, the absence of key natural airborne aerosol formation pathways in the




model, potentially involving iodine oxoacids, organic vapours, or their interactions with $H_2SO_4$, also contribute to both the preindustrial and present day CCN populations in the atmosphere. These factors could skew estimates of preindustrial aerosol baselines and the strength of aerosol-cloud interactions.

Our results underscore the need to incorporate comprehensive halogen chemistry, especially iodine-related aerosol nucleation mechanisms, which are increasingly recognised in both laboratory and field studies as major contributors to marine aerosol formation. Enhancing model treatments of organic aerosol sources and their oxidation pathways is also essential, particularly for the upper troposphere, where long range transport of isoprene-derived aerosols can shape aerosol populations on a global scale.

Looking ahead, model improvement efforts should prioritise process-based improvement over empirical tuning. The long-lasting uncertainties in aerosol forcing, despite decades of model development, highlight the limitations of ad-hoc adjustments and point to the importance of mechanistic understanding. Capturing the underlying physics and chemistry of aerosol processes is one of the prerequisite for building models capable of producing robust climate projections and informing effective mitigation strategies.

The ATom dataset offers a powerful benchmark for advancing this work, but fully leveraging its value will require sustained investment in experimental research. Laboratory studies of nucleation chemistry and targeted field campaigns are particularly needed to resolve the complex pathways governing natural aerosol formation. As anthropogenic sulfur emissions continue to decline, accurately representing these natural processes will become even more crucial for predicting future climate trajectories.

*Data availability.* The ATom aerosol number size distribution measurements are publicly accessible at https://doi.org/10.3334/ORNLDAAC/2111
(Brock et al., 2022). The ATom atmospheric trace gas measurements are archived at https://doi.org/10.3334/ORNLDAAC/1925 (Wofsy et al., 2021). ATom aerosol composition measurements are also available at https://doi.org/10.3334/ORNLDAAC/1925 (Wofsy et al., 2021); however, the specific dataset utilised in this study, together with the real-time transmission correction curve, was provided by P.C.-J. and J.L.J. The integrated 5-minute resolution data product combining collocated UKESM1.1 model output with subsampled ATom observational data will be deposited in the Zenodo repository upon acceptance of this manuscript. Additional UKESM1.1 model output is available from
X.-C.H. upon reasonable request.

*Author contributions.* X.-C.H., N.L.A., A.T.A., and H.G. conceived this study. N.L.A. and A.T.A. provided the computational environment and resources on ARCHER2 and H.G. on Bridges-2. X.-C.H., H.D., X.W., J.W., and H.G. carried out model development and module incorporation. X.-C.H. carried out model simulations with assistance from N.L.A., J.W., and H.G. Additionally, P.C.-J., B.N., A.K., D.B., J.L.J., and C.J.W. provided ATom observational data. X.-C.H. carried out data analysis and wrote the manuscript. All authors have commented
and edited the manuscript.

*Competing interests.* At least one of the (co-)authors is a member of the editorial board of *Atmospheric Chemistry and Physics*.



*Acknowledgements.* This work used the ARCHER2 UK National Supercomputing Service. This work used the Bridges-2 machine at the Pittsburgh Supercomputing Center (PSC) through allocation atm200005p from the Advanced Cyberinfrastructure Coordination Ecosystem: Services and Support (ACCESS) program, which is supported by National Science Foundation grants no. 2138259, 2138286, 2138307, 2137603, and 2138296. We additionally thank Charles A. Brock, William H. Brune, Thomas B. Ryerson, Alexander B. Thames, Chelsea R. Thompson, and Steven C. Wofsy for providing data directly relevant to this study, and extend our gratitude to the entire ATom team for their extraordinary work in collecting and curating these invaluable atmospheric measurements.

*Financial support.* X.-C.H. acknowledges support from Research Council of Finland grant no. 359331, 349659, 371185. H.G. and H.D. acknowledge support from the NASA ROSES program via grant no. 80NSSC19K0949, and H.G. additionally acknowledges support from the US Department of Energy Atmospheric System Research program via grant no. DE-SC-0022227. A.T.A. acknowledges Natural Environment Research Council CARES project no. NE/W009412/1. N.L.A. and M.R.R. thank Natural Environment Research Council for financial support through the VISION project, NE/Z503393/1. D.B. acknowledges support from NASA grant no. NNX15AJ91A. P.C.-J., B.N., and J.L.J. are supported by NASA grant no. 80NSSC21K1451 and 80NSSC23K0828.



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



**Appendix A**

**Table A1.** Model-to-ATom median ratios for environmental conditions, precursor vapours, aerosol number size distributions, and chemical composition between 2 - 8 km altitude. Ratios are calculated using all available ATom data between 2 - 8 km for comparison with different UKESM1.1 configurations implementing various SA (sulfuric acid)-based nucleation schemes (see Table 2 for simulation details). The median values are calculated from the point-by-point model-to-ATom ratios (i.e. the ratio is computed at each location or time point, followed by taking the median), and therefore do not necessarily equal the ratio of the median model value to the median ATom value. Values greater than 1 indicate model overestimation, while values less than 1 indicate model underestimation relative to ATom observations. Environmental conditions include T, RH, and CS of dry aerosols. Precursor vapours include DMS, $SO_2$, $NH_3$, $O_3$, and OH. Aerosol number concentrations are reported for nucleation (dry diameter < 10 nm), Aitken (10 - 100 nm), accumulation (100 - 1000 nm), and coarse (1000 - 10,000 nm) modes, along with total number concentration. Chemical composition includes sulfate, organic matter, ammonium, nitrate, and seasalt mass concentrations, with total mass representing their sum. Missing values indicate that ammonium and nitrate components are not included in simulations without the ammonium nitrate scheme (SA-$H_2O$[default] and SA-$NH_3$-noNit). The extreme values in the $NH_3$ ratios result from extremely low $NH_3$ concentrations inferred from aerosol acidity, as $NH_3$ is virtually absent in the specified region.

| | $H_2SO_4$(SA)-based schemes | | | | | | | | |
| --- | --- | --- | --- | --- | --- | --- | --- | --- | --- |
| | $H_2O$(default) | $NH_3$-noNit | $NH_3$-slow | $NH_3$-Lana | $NH_3$(benchmark) | $NH_3$-CS2 | $NH_3$-MSA | $NH_3$-IPSOA | $NH_3$-IPSOA×10 |
| **Temperature** | 1.00 | 1.00 | 1.00 | 1.00 | 1.00 | 1.00 | 1.00 | 1.00 | 1.00 |
| **RH** | 1.10 | 1.10 | 1.10 | 1.10 | 1.10 | 1.09 | 1.10 | 1.09 | 1.10 |
| **CS dry** | 1.26 | 1.35 | 1.40 | 1.50 | 1.41 | 1.31 | 1.46 | 1.42 | 1.53 |
| **DMS** | 0.44 | 0.43 | 0.62 | 0.94 | 0.63 | 0.54 | 0.63 | 0.64 | 0.66 |
| **$SO_2$** | 1.02 | 1.02 | 1.12 | 1.15 | 1.16 | 0.47 | 1.14 | 1.15 | 1.11 |
| **$NH_3$** | 2797188.44 | 2846798.75 | 33.66 | 1.66 | 7.91 | 38.70 | 3.03 | 7.24 | 6.01 |
| **$O_3$** | 1.09 | 1.08 | 1.03 | 1.01 | 1.02 | 0.99 | 1.02 | 1.02 | 1.01 |
| **OH** | 1.13 | 1.13 | 1.06 | 1.04 | 1.05 | 1.08 | 1.06 | 1.06 | 1.05 |
| **Nucleation** | 0.92 | 1.91 | 0.57 | 0.30 | 0.37 | 0.22 | 0.21 | 0.35 | 0.27 |
| **Aitken** | 1.18 | 1.49 | 1.07 | 0.94 | 0.91 | 0.77 | 0.94 | 0.92 | 0.83 |
| **Accumulation** | 0.77 | 0.83 | 0.92 | 1.00 | 0.96 | 0.97 | 0.97 | 0.98 | 1.10 |
| **Coarse** | 1.30 | 1.28 | 1.43 | 1.50 | 1.51 | 1.47 | 1.50 | 1.46 | 1.65 |
| **Total number** | 1.03 | 1.39 | 0.87 | 0.74 | 0.74 | 0.63 | 0.70 | 0.73 | 0.67 |
| **Sulfate** | 1.42 | 1.48 | 1.37 | 1.46 | 1.32 | 1.06 | 1.45 | 1.35 | 1.29 |
| **Organic** | 0.95 | 1.01 | 0.96 | 0.93 | 0.93 | 0.82 | 0.89 | 0.95 | 1.52 |
| **Ammonium** | – | – | 7.47 | 7.97 | 7.68 | 7.04 | 7.88 | 7.77 | 7.34 |
| **Nitrate** | – | – | 4.48 | 8.90 | 9.82 | 8.33 | 9.54 | 9.54 | 9.10 |
| **Seasalt** | 2.10 | 2.19 | 0.66 | 0.72 | 0.75 | 0.74 | 0.70 | 0.73 | 0.70 |
| **Total mass** | 1.26 | 1.29 | 1.37 | 1.46 | 1.36 | 1.23 | 1.43 | 1.38 | 1.55 |





**Table A2.** Model-to-ATom median ratios for environmental conditions, precursor vapours, aerosol number size distributions, and chemical composition in upper troposphere (8 - 12 km altitude). Ratios are calculated using all available ATom data between 8 - 12 km for comparison with different UKESM1.1 configurations implementing various SA (sulfuric acid)-based nucleation schemes (see Table 2 for simulation details). The median values are calculated from the point-by-point model-to-ATom ratios (i.e. the ratio is computed at each location or time point, followed by taking the median), and therefore do not necessarily equal the ratio of the median model value to the median ATom value. Values greater than 1 indicate model overestimation, while values less than 1 indicate model underestimation relative to ATom observations. Environmental conditions include T, RH, and CS of dry aerosols. Precursor vapours include DMS, $SO_2$, $NH_3$, $O_3$, and OH. Aerosol number concentrations are reported for nucleation (dry diameter < 10 nm), Aitken (10 - 100 nm), accumulation (100 - 1000 nm), and coarse (1000 - 10,000 nm) modes, along with total number concentration. Chemical composition includes sulfate, organic matter, ammonium, nitrate, and seasalt mass concentrations, with total mass representing their sum. Missing values indicate that ammonium and nitrate components are not included in simulations without the ammonium nitrate scheme (SA-$H_2O$[default] and SA-$NH_3$-noNit). The extreme values in the $NH_3$ ratios result from extremely low $NH_3$ concentrations inferred from aerosol acidity, as $NH_3$ is virtually absent in the specified region.

| | **$H_2SO_4$(SA)-based schemes** | | | | | | | | |
| | $H_2O$(default) | $NH_3$-noNit | $NH_3$-slow | $NH_3$-Lana | $NH_3$(benchmark) | $NH_3$-CS2 | $NH_3$-MSA | $NH_3$-IPSOA | $NH_3$-IPSOA×10 |
|---|---|---|---|---|---|---|---|---|---|
| **Temperature** | 1.00 | 1.00 | 1.00 | 1.00 | 1.00 | 1.00 | 1.00 | 1.00 | 1.00 |
| **RH** | 1.15 | 1.14 | 1.15 | 1.15 | 1.15 | 1.13 | 1.15 | 1.15 | 1.15 |
| **CS dry** | 1.07 | 1.10 | 1.10 | 1.20 | 1.10 | 0.93 | 1.14 | 1.11 | 1.25 |
| **DMS** | 0.16 | 0.17 | 0.21 | 0.18 | 0.23 | 0.40 | 0.23 | 0.26 | 0.37 |
| **$SO_2$** | 0.76 | 0.75 | 0.81 | 0.88 | 0.83 | 0.47 | 0.82 | 0.80 | 0.80 |
| **$NH_3$** | 3644293052.67 | 3888048993.51 | 30685.18 | 0.00 | 0.03 | 0.77 | 0.05 | 0.02 | 0.02 |
| **$O_3$** | 1.21 | 1.20 | 1.15 | 1.14 | 1.13 | 1.07 | 1.14 | 1.14 | 1.13 |
| **OH** | 1.09 | 1.08 | 1.01 | 1.00 | 0.99 | 0.99 | 0.99 | 1.00 | 0.99 |
| **Nucleation** | 6.30 | 8.87 | 3.30 | 1.74 | 1.79 | 1.37 | 1.42 | 1.72 | 1.24 |
| **Aitken** | 1.88 | 2.11 | 1.64 | 1.51 | 1.39 | 1.01 | 1.51 | 1.36 | 1.22 |
| **Accumulation** | 0.39 | 0.42 | 0.46 | 0.57 | 0.52 | 0.50 | 0.50 | 0.55 | 0.70 |
| **Coarse** | 1.20 | 1.13 | 1.23 | 1.27 | 1.31 | 1.28 | 1.29 | 1.24 | 1.36 |
| **Total number** | 2.11 | 2.58 | 1.51 | 1.14 | 1.11 | 0.85 | 1.08 | 1.08 | 0.94 |
| **Sulfate** | 1.09 | 1.10 | 1.04 | 1.20 | 1.01 | 0.73 | 1.13 | 1.02 | 0.99 |
| **Organic** | 0.43 | 0.45 | 0.44 | 0.45 | 0.44 | 0.46 | 0.43 | 0.49 | 0.86 |
| **Ammonium** | – | – | 8.47 | 9.70 | 8.63 | 7.44 | 9.42 | 8.68 | 8.40 |
| **Nitrate** | – | – | 0.64 | 0.46 | 0.93 | 2.73 | 0.42 | 1.06 | 0.93 |
| **Seasalt** | 0.26 | 0.28 | 0.13 | 0.15 | 0.15 | 0.15 | 0.14 | 0.15 | 0.13 |
| **Total mass** | 0.72 | 0.73 | 0.80 | 0.94 | 0.83 | 0.77 | 0.87 | 0.86 | 1.01 |





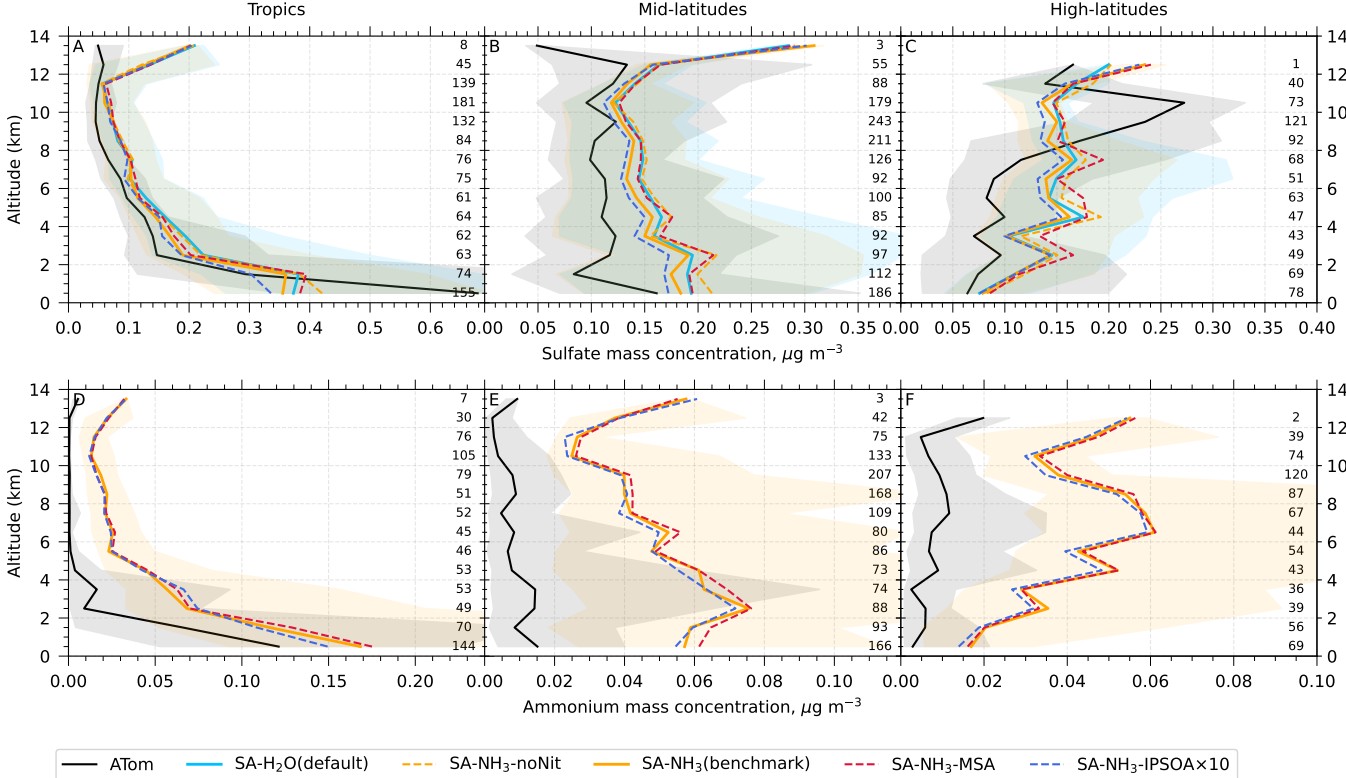

**Figure A1.** Vertical profiles of sulfate and ammonium mass concentrations in particles smaller than 1 $\mu$m ($\mu$g m$^{-3}$). ATom observations (black lines) and selected model simulations (coloured lines; see Table 2 for simulation details) are presented. Median values are shown for sulfate mass concentrations in the tropics (A, 25 °S to 25 °N), mid-latitudes (B, 25 - 60 °N/S), and high-latitudes (C, 60 - 90 °N/S), and for ammonium mass concentrations in the tropics (D), mid-latitudes (E), and high-latitudes (F). Shaded areas for solid lines represent the interquartile range (25th to 75th percentiles) for both ATom observations and model simulations, indicating the variability in aerosol concentrations. Numbers on the right edge of each panel show the sample size used to calculate median values for each altitude bin at 5-minute temporal resolution.



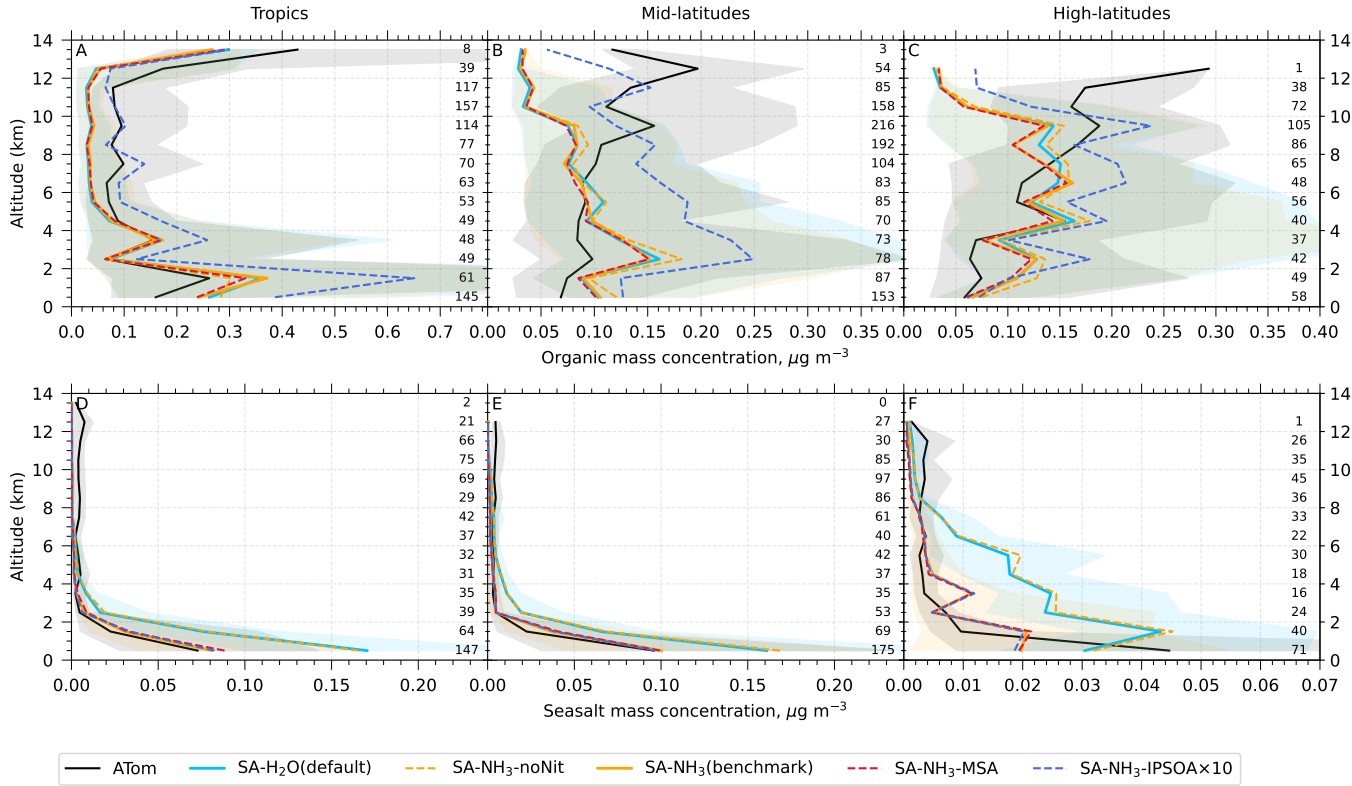

**Figure A2.** Vertical profiles of organic and seasalt mass concentrations in particles smaller than 1 $\mu$m ($\mu$g m$^{-3}$). ATom observations (black lines) and selected model simulations (coloured lines; see Table 2 for simulation details) are presented. Median values are shown for organic mass concentrations in the tropics (A, 25 °S to 25 °N), mid-latitudes (B, 25 - 60 °N/S), and high-latitudes (C, 60 - 90 °N/S), and for seasalt mass concentrations in the tropics (D), mid-latitudes (E), and high-latitudes (F). Shaded areas for solid lines represent the interquartile range (25th to 75th percentiles) for both ATom observations and model simulations, indicating the variability in aerosol concentrations. Numbers on the right edge of each panel show the sample size used to calculate median values for each altitude bin at 5-minute temporal resolution.