# Peer review of "Evaluation of UKESM aerosol size and composition using ATom measurements indicates missing marine aerosol formation mechanisms"

_EGUsphere, 2025_

## Author Comment (AC1)

**Reviewer 1**

The manuscript evaluates UKESM1.1 against ATom aircraft observations and implements several process upgrades ($H_2SO_4$–$NH_3$ nucleation based on Dunne et al., MSA condensation, ammonium nitrate scheme, alternative DMS chemistry/ climatology, isoprene SOA scheme). The paper identifies a persistent mismatch: MBL nucleation/Aitken aerosols are underpredicted while precursors are overpredicted, and UT nucleation is sometimes overpredicted but growth to accumulation mode is insufficient. The topic is important and the model-development tests are useful. Overall, the study is well-designed and easy to follow, with most conclusions supported by model results. Therefore, I think the paper may be accepted for publication although I'd like the authors to consider some options to improve their work and polish the manuscript.

We thank the reviewer for their constructive and insightful comments. The reviewer's comments are shown below in black, our responses are provided in blue, and the corresponding revisions in the manuscript are highlighted in orange.

1. What my concern is the text's length and narrative structure. For instance, the Result section's current variable category organization results in a long and detailed narrative that makes it difficult for the reader to quickly identify and connect the most important findings. Consequently, the significance of the paper's significant findings might be lessened. I suggest moving more technical or secondary information to the Supplement and reorganizing this section to highlight the most important findings.

We thank the reviewer for this constructive suggestion regarding the length and narrative structure of the manuscript. We agree that the manuscript is relatively long. We deliberately structured it to accommodate both readers who wish to focus on specific parameters and those who aim to obtain a comprehensive understanding of the overall model performance.

With this objective in mind, the Results section was organised in a parameter-based manner, functioning as a reference-style overview of all examined variables, while the Discussion section provides a synthesis of the results with particular emphasis on processes in the marine boundary layer and the upper troposphere. To ensure clarity and internal consistency, each parameter is

discussed independently in the Results section, which may lead to some unavoidable minor repetition.

Nevertheless, we agree with the reviewer that further shortening and streamlining would improve readability. Accordingly, we have condensed several parts of the Results section. We decided not to include supplementary material, as it is unnecessary and would potentially dilute the coherence of the self-consistent information already presented in the manuscript. In addition, we have added a guiding paragraph at the end of the Introduction to better orient readers to the manuscript structure and to highlight the key findings.

Lines 133 - 138: Consistent with this process-oriented approach, the manuscript is structured as follows. The Methods section provides an overview of UKESM1.1, including its aerosol and chemistry modules, as well as the model developments and implementations carried out in this study. The Results section is organised by individual parameters, allowing readers to focus on specific variables of interest. The Discussion section synthesises these results and highlights their implications for aerosol processes in the marine boundary layer and the upper troposphere. Finally, we present recommendations for further development of UKESM to improve the representation of aerosols and cloud condensation nuclei.

2. L189 The authors stated that anthropogenic emissions are emitted into the lowest model. What is the height of the surface layer?

The lowest grid cell (surface layer) extends up to 35 metres in the model. This information has been added to the manuscript.

Lines 216 - 218: $SO_2$ emitted from both the energy and industrial sectors is released into the model's surface layer (up to 35 m), consistent with the treatment of other trace gas emissions in UKCA.

3. The authors should precisely specify model setups, including simulation periods, initial and boundary conditions, and details regarding the ERA5 dataset.

We made the following changes to specify the model setups and runs, in addition to the existing ones:

Lines 157 - 158: Surface concentrations of methane ($CH_4$), nitrous oxide ($N_2O$) and other well-mixed greenhouse gases were prescribed following the SSP3-7.0 projection.

Additionally, we have now consolidated information from other sections into Section 2.5 to provide a clear and explicit description of the model simulations and the ERA5 dataset used.

Lines 485 - 498: The simulations in this study employ a horizontal resolution of 1.875◦ longitude × 1.25◦ latitude, corresponding to approximately 135 km at the equator. Vertically, the model utilises 85 levels extending up to 85 km from the Earth's surface, with 50 levels concentrated between 0 and 18 km, which are the primary region of focus for this study.

The model simulations are nudged using horizontal wind and temperature fields from the ERA5 reanalysis (Telford et al., 2008; Dee et al., 2011; Hersbach et al., 2020) corresponding to the period of ATom observation. This nudging aims to reproduce the specific meteorological conditions at the time and location of the measurements, thereby reducing model biases often present in free-running model configurations (Kipling et al., 2013). Since the temporal resolution of the ERA5 reanalysis is 6 hours, the relaxation time constant for the nudged simulations is set to 6 hours. Nudging is applied vertically between model levels 12 and 80.

To cover the full ATom observation period, the model was run from July 2016 to June 2018, with a six-month spin-up (initialisation) period from January to June 2016. For comparison with ATom observations, model outputs are retrieved as instantaneous values at a high temporal resolution of one hour. This high frequency is intended to minimise sampling bias from the model. However, due to the substantial disk space required for these high-resolution outputs (roughly 25 Gigabytes per day), model data are saved only for the specific dates corresponding to ATom flights for subsequent offline analysis.

4. In section 2.3.1, the authors enumerated four nucleation schemes derived from CERN CLOUD (Cosmics Leaving Outdoor Droplets) studies, which encompassed binary and ternary ion-induced nucleation. Are the four schemes examined in this study? Please clarify. To prevent ambiguity, the authors should provide a detailed description of the nucleation methods employed in the model within the text or appendix, as the objective of the study is to investigate aerosol formation mechanisms.

The sulfuric acid-ammonia nucleation mechanism implemented in UKESM1.1 follows the parameterisation of Dunne et al. (Dunne et al., 2016) and consists of four individually parameterised sub-schemes: (1) binary neutral $H_2SO_4$-$H_2O$ nucleation, (2) binary ion-induced $H_2SO_4$-$H_2O$ nucleation, (3) ternary neutral $H_2SO_4$-$NH_3$-$H_2O$ nucleation, and (4) ternary ion-induced $H_2SO_4$-$NH_3$-$H_2O$ nucleation. All four sub-schemes are included and active in the simulations presented in this study and are collectively referred to as the SA-$NH_3$ ($H_2O$) nucleation mechanism in UKESM1.1. As the formulation of these nucleation parameterisations is extensive and has been comprehensively documented in Dunne et al. (2016) and Gordon et al. (Gordon et al., 2017), we provide a concise summary of their key features and refer the reader to the original study for full methodological details in section 2.3.1. In order to avoid ambiguity, we modified the relevant part to:

Lines 310 - 319: Experiments reported in Dunne et al. (2016) identified four distinct aerosol nucleation regimes, differentiated by the concentrations of $NH_3$ and the presence of atmospheric ions. The first regime is termed binary neutral nucleation of $H_2SO_4$-$H_2O$; this represents the experimental version of the theoretical nucleation scheme used by default in UKESM1.1 (Kulmala et al., 1998; Vehkamäki et al., 2002). This regime's rate depends only on the concentrations of $H_2SO_4$ and $H_2O$. The second regime is termed binary ion-induced nucleation of $H_2SO_4$-$H_2O$, describing nucleation in the presence of atmospheric ions. Consequently, the nucleation rate of this regime exhibits an additional dependence on the concentration of atmospheric ions. The third regime is ternary neutral nucleation of $H_2SO_4$-$NH_3$(-$H_2O$), which depends on the concentrations of $H_2SO_4$, $NH_3$, and $H_2O$. As $H_2O$ participation is fundamental to atmospheric nucleation, it is often implicitly assumed and omitted from the nomenclature. Finally, the fourth regime is ternary ion-induced nucleation of $H_2SO_4$-$NH_3$, which incorporates the enhancing effect of atmospheric ions on the ternary system.

5. It is preferable for readers to have a description of how the $K_{RH}$ adapts the nucleation process. The authors modified the original formula ($K_{RH} = 1 + c1(RH - 0.38) + c2(RH - 0.38)^3(T - 208)^2$, where $c1 = 1.5 \pm 1.3$, $c2 = 0.045 \pm 0.003$) to $K_{RH} = (RH/0.38) \times [1 + 0.02 \times (RH - 0.38) \times (T - 208)^{1.2}]$. The alteration is a significant adjustment that directly affects nucleation rates in the current study. To substantiate this, kindly furnish the precise algebraic expression, parameter values, and rationale for the selected coefficients (e.g., exponent 1.2, coefficient 0.02) inside the text or the appendix.

We agree with the reviewer that the modification to the RH enhancement factor represents a substantial change that directly affects nucleation rates. After further investigation, we concluded that including this modification in the present manuscript is premature. First, a more careful assessment revealed that the available experimental constraints are not sufficiently robust to justify deviation from the published parameterisation. Second, introducing and justifying a revised formulation would distract from the primary scope of this already extensive, model-focused study, which is not intended to propose or validate new nucleation parameterisations.

Accordingly, we have re-run all simulations using the original, published RH enhancement parameterisation and now present results based exclusively on the original formulation. We anticipate that additional dedicated chamber measurements will enable a more rigorous assessment of the RH dependence of nucleation in a future, focused study. As a result, the relevant discussions have been modified as below:

Lines 320 - 324: The impact of $H_2O$ on aerosol nucleation is represented as a relative humidity dependent multiplier ($K_{RH}$) applied to the nucleation rate calculated based on $H_2SO_4$, $NH_3$, and atmospheric ion concentrations. The formula reads $K_{RH} = 1 + c1(RH - 0.38) + c2(RH - 0.38)^3(T - 208)^2$, where $c1 = 1.5 \pm 1.3$, $c2 = 0.045 \pm 0.003$, RH is the relative humidity expressed as a fraction, and T is the temperature in Kelvin. This formulation uses 38 % RH (RH = 0.38) as the reference condition, reflecting the humidity level at which most of the underlying experiments were performed (Dunne et al., 2016).

All figures, tables, and numerical values in the manuscript have been updated to reflect this change. Importantly, this revision does not alter the overall conclusions of this study.

6. The authors emphasized that the CS2 scheme employed distinct emission inventories compared to the Strat-Trop chemistry scheme. What are the disparities in essential pollutants that regulate nucleation, such as $SO_2$ and precursors influencing OH radicals (e.g. NOx, VOCs)?

In the original manuscript, the CS2 scheme used 2010 time-slice $SO_2$ and $NO_x$ emissions, along with other biogenic emissions, consistent with the original studies (Archer-Nicholls et al., 2021; Weber et al., 2021). In contrast, the ST scheme used $SO_2$ and $NO_x$ emissions from the SSP3-7.0 scenario for the time period 2016 - 2018 of the simulations. This design choice was intended to

enable direct comparisons between the ST and CS2 schemes while maintaining consistency with their respective original emission inventories.

However, as pointed out by the reviewer, the inconsistencies between the emissions directly affecting aerosol nucleation might have major impacts on aerosol distribution. Therefore, we decided to implement $SO_2$, NO, organic carbon, black carbon, ammonia, methane and carbon monoxide emissions from the SSP3-7.0 scenario to the CS2 simulations while keeping other organic emissions the same with the original studies, mostly 2010 time-slice (Archer-Nicholls et al., 2021; Weber et al., 2021). This modification serves to remove the impact from the inconsistent anthropogenic $SO_2$, $NO_x$ and other anthropogenic emissions. Therefore, the comparison between CS2 and ST presented in this study focuses on comparing the different chemistry schemes of CS2 and ST, as well as the impact of the additional VOCs in CS.

Under present-day conditions, CS2 includes approximately 45 TgC/yr additional anthropogenic VOC emissions and 15 TgC/yr additional biomass-burning VOC emissions compared to ST (Archer-Nicholls et al., 2021; Weber et al., 2021). Because CS2 provides a more detailed representation of monoterpene and isoprene chemistry and their atmospheric fates, the resulting oxidant fields, particularly $O_3$ and OH, differ between the two schemes. In addition, as summarised in Table 1 of the paper, the DMS chemistry is treated differently in ST and CS2, which in turn leads to differences in the chemical production of $SO_2$ within UKESM.

Lines 389 - 396: For example, most anthropogenic and biogenic emissions are based on datasets spanning 2000 - 2010 or on a 2010 time-slice. In this study, we updated selected anthropogenic emissions — including $SO_2$, nitrogen monoxide, organic carbon, black carbon, ammonia, methane, and carbon monoxide - to use CMIP6 SSP3-7.0 emissions to be consistent with the ST simulations. Emissions of volatile organic compounds (VOCs), most of which are unique in CS2, are left unchanged and therefore remain consistent with the original implementations described by Archer-Nicholls et al. (2021) and Weber et al. (2021). This modification removes the influence of inconsistent anthropogenic emissions, such that the comparison between CS2 and ST presented in this study primarily reflects differences in the underlying chemistry schemes, as well as the impact of the additional VOCs included in CS2.

Figure R2 shows the $SO_2$ (panel A) and OH (panel B) mixing ratios from the SA-NH$_3$ simulation using ST chemistry, together with the ratios of $SO_2$ (panel C) and OH (panel D) between the SA-NH$_3$-CS2 and SA-NH$_3$ simulations. Overall, the CS2 scheme produces lower $SO_2$ concentrations globally, except over Northern Hemisphere continental regions. OH concentrations are generally higher in Northern Hemisphere oceans in CS2 but lower in the Southern Hemisphere oceans. A more comprehensive comparison of the CS2 and ST chemistry schemes is provided in previous

studies (Archer-Nicholls et al., 2021; Weber et al., 2021). As detailed comparison of the CS2 and ST schemes is beyond the scope of the current study, we only focus on their general performance in this study.

[Figure]

*Figure R 1. Global distribution of marine boundary layer SO₂ and OH following implementation of the CS2 scheme. Panels show vapour concentrations in the marine boundary layer (0 - 2 km altitude) averaged over monthly means from July 2016 to June 2018: (A) SO₂ from the benchmark SA-NH₃ simulation, (B) OH from the benchmark SA-NH₃ simulation, (C) ratio of SO₂ between SA-NH₃-CS2 and benchmark SA-NH₃ simulations, and (D) ratio of OH between SA-NH₃-CS2 and benchmark SA-NH₃ simulations.*

**Reviewer 2**

The evaluation of the United Kingdom Earth System Model version 1.1 (UKESM1.1) using ATom observations highlights significant biases in model gas concentrations and aerosol size distributions. The study brings up the potential sources of these biases in a well-structured way, evaluating new approaches for UKESM aerosol and chemistry from incorporating new Ammonia-based nucleation pathways, different DMS chemistry and climatology, IPSOA and ammonium nitrate schemes, and MSA condensation. The claims made by the authors are displayed and answered sufficiently. In my assessment, the methodology, the data are clearly presented, and the conclusions are well-supported. Overall, the manuscript makes a meaningful contribution to the development of UKESM and ESM in general, and I believe it meets the standards for publication. Therefore, I think this paper should be accepted.

**The review below is structured as direct responses to the questions outlined in ACP's review criteria.**

- **Does the paper address relevant scientific questions within the scope of ACP?**

This paper examines how the aerosol lifecycle, including precursor vapors, number size distributions, and chemical composition, can be represented in global models, as well as how these representations can be evaluated, of which fall within the scope of ACP.

- **Does the paper present novel concepts, ideas, tools, or data?**

The paper primarily focuses on evaluating and improving existing elements within the UKESM1.1 model, rather than presenting entirely novel concepts or tools. The paper implements and tests several updated or alternative process representations within UKESM1.1. The paper identifies missing parameterizations likely necessary to resolve existing model biases.

The study utilizes global-scale aircraft observations from the Atmospheric Tomography (ATom) mission as a comprehensive dataset for evaluation. The paper presents novel model outputs to compare and evaluate towards this ATom dataset.

- **Are substantial conclusions reached?**

Substantial conclusions are reached regarding the need to include representation of currently missing aerosol sources and updated chemistry schemes in climate models like UKESM1.1 to accurately simulate the remote marine aerosol lifecycle.

We thank the reviewer for their positive and thoughtful assessment of our manuscript. We appreciate their recognition of the relevance, clarity, and significance of this work, and we are grateful for the constructive comments provided below, which have helped us further improve the manuscript. We respond to these points individually below.

- **Are the scientific methods and assumptions valid and clearly outlined?**

Some weaknesses in the methodology are outlined below with questions. If no question is asked, these issues are adequately addressed in the paper.

The SOA formation is represented by scaling monoterpene oxidation yield (by a factor of two) to implicitly account for other sources like isoprene. This scaling factor (as suggested in the paper) is highly uncertain. Which aerosol size mode (nucleation, Aitken, accumulation, or coarse) is the SOA mass yield formed into?

The stated MSA condensation scheme is "preliminary" and stated to "not yet be fully validated" and act as "Initial proof-of-concept implementation". The merging of MSA as H2SO4 aerosol mass also does not account for the optical properties of MSA in the particle phase in this new scheme.

We agree with the reviewer that the default UKESM scaling factor applied to monoterpene SOA to account for missing isoprene SOA is not ideal. This limitation has been addressed in recent model developments that introduce explicit isoprene emissions and chemistry (Weber et al., 2022), as implemented in the simulations denoted SA-NH$_3$-IPSOA and SA-NH$_3$-IPSOA×10. In all model simulations, condensable organic vapours are assumed to condense onto all particle modes (nucleation, Aitken, accumulation, and coarse). We added the text below to explicitly explain this:

Lines 420 - 421: After formation, the condensable oxidation products of monoterpenes and isoprene are assumed to condense irreversibly onto all particle modes, including nucleation, Aitken, accumulation, and coarse.

The current implementation of the MSA condensation scheme is preliminary, primarily because of the lack of experimental data to constrain the condensation rate, and because gas-phase MSA production from DMS has not yet been fully quantified and implemented. Addressing these experimental uncertainties requires future studies; once such data become available, the present MSA condensation scheme can be readily updated to incorporate the new findings.

The reviewer is correct that MSA and SA are currently merged in the aerosol phase as "sulfate", despite their potentially different optical properties in the atmosphere. As this study focuses on evaluating aerosol number and mass, rather than aerosol optical properties or climate impacts, we believe that this simplification does not affect the conclusions. A dedicated MSA aerosol component (and its optical properties) will be developed in the future if separate treatment of MSA and SA becomes necessary.

- **Are the results sufficient to support the interpretations and conclusions?**

In summary, the detailed comparison against the unique ATom dataset and the application of process-specific sensitivity tests provide sufficient evidence to back the paper's primary conclusions concerning the need to implement new mechanisms. Adequately addressing the weaknesses of the implementations presented in this study.

I would like to see a short mention of the limitation imposed by the computational time required by the chemical solver, if the suggested pathways were implemented, given that most ESMs are constrained in this regard. It would also then be helpful to clarify whether these potential future model implementations are intended for the CMIP version or for UKESM.

As required by the reviewer, we added the below text to highlight that additional mechanisms would require additional computational time. We also highlight that the need of these newly implemented mechanisms to constrain marine aerosols and their climate impacts.

Lines 1044 - 1050: It should be noted that the need to implement new aerosol formation schemes is not unique to UKESM; other Earth system model evaluation studies, such as those using EC-Earth3, have reached similar conclusion (Svenhag et al., 2024). While implementing these comprehensive chemical and microphysical pathways will increase the computational demand on the model's chemical solver, such developments are essential for capturing the true sensitivity of the climate to aerosols. Once these full mechanisms are implemented and validated, they can be systematically simplified or parameterised based on their performance to reduce the

computational burden for long-term climate simulations without sacrificing significant accuracy, for the eventual CMIP implementations.

The recommendations outlined in this study are primarily intended for the development of the next generation of UKESM. While some of the more mature parameterisations could eventually be integrated into the CMIP configurations, the initial focus remains on improving the process-level fidelity within the UKESM framework to better understand aerosol-climate feedbacks.

- **Is the description of experiments and calculations sufficiently complete and precise to allow their reproduction by fellow scientists (traceability of results)?**

Yes.

- **Do the authors give proper credit to related work and clearly indicate their own new/original contribution?**

Yes.

- **Does the title clearly reflect the contents of the paper?**

Yes.

- **Does the abstract provide a concise and complete summary?**

Yes.

- **Is the overall presentation well structured and clear?**

Yes.

- **Is the language fluent and precise?**

Yes.

- **Are mathematical formulae, symbols, abbreviations, and units correctly defined and used?**

Yes.

- **Should any parts of the paper (text, formulae, figures, tables) be clarified, reduced, combined, or eliminated?**

Overall, the authors require the reader to frequently jump between figures to follow and support the scientific conclusions they present. Some figures could perhaps be moved to a supplementary section, or minor adjustments could be made to the various curtain plots of the UKESM experiments, as differences are difficult to discern in most figures. However, the overall structure of the figures is visually acceptable.

We thank the reviewer for this comment. We agree that, due to the evaluation of numerous aerosol chemical and microphysical parameters across up to nine simulations, readers may occasionally need to refer to multiple figures. We have made efforts to minimise redundancy wherever possible. For example, in the curtain plots, we only present the essential model simulations for each parameter type - environmental conditions, vapours, aerosol number, and mass - while omitting less critical plots. This approach allows us to maintain clarity and focus while still providing the necessary information to support our conclusions.

We have re-evaluated all figures and decided that the original Figure 9 can be moved to the Appendix as the new Figure A1. We retained all other figures since they all provide necessary information for the main conclusions in the main text. We have also made additional efforts to streamline the Results section and further reduce redundancy to enhance readability.

**Are the number and quality of references appropriate?**

In general, yes. The cited findings from previous literature identify candidate mechanisms for implementation. Moreover, claims regarding insufficient model oxidation products and the need for new mechanistic pathways could be further supported by results from the ESM study on EC-Earth3 by Svenhag et al. (2024), which used TM5 to generate ion-dependent $NH_3$–$H_2SO_4$ nucleation via a lookup-table approach.

We thank the reviewer for pointing out the relevant EC-Earth3 development. We have now included a discussion of Svenhag et al. (Svenhag et al., 2024) in the revised manuscript, as outlined below.

Lines 1044 - 1045: It should be noted that the need to implement new aerosol formation schemes is not unique to UKESM; other Earth system model evaluation studies, such as those using EC-Earth3, have reached similar conclusion (Svenhag et al., 2024).

- **Is the amount and quality of supplementary material appropriate?**

Yes.

References:

Archer-Nicholls, S., Abraham, N. L., Shin, Y. M., Weber, J., Russo, M. R., Lowe, D., Utembe, S. R., O'Connor, F. M., Kerridge, B., Latter, B., Siddans, R., Jenkin, M., Wild, O., and Archibald, A. T.: The Common Representative Intermediates Mechanism Version 2 in the United Kingdom Chemistry and Aerosols Model, J Adv Model Earth Syst, 13, e2020MS002420, https://doi.org/10.1029/2020MS002420, 2021.

Dunne, E. M., Gordon, H., Kürten, A., Almeida, J., Duplissy, J., Williamson, C., Ortega, I. K., Pringle, K. J., Adamov, A., Baltensperger, U., Barmet, P., Benduhn, F., Bianchi, F., Breitenlechner, M., Clarke, A., Curtius, J., Dommen, J., Donahue, N. M., Ehrhart, S., Flagan, R. C., Franchin, A., Guida, R., Hakala, J., Hansel, A., Heinritzi, M., Jokinen, T., Kangasluoma, J., Kirkby, J., Kulmala, M., Kupc, A., Lawler, M. J., Lehtipalo, K., Makhmutov, V., Mann, G., Mathot, S., Merikanto, J., Miettinen, P., Nenes, A., Onnela, A., Rap, A., Reddington, C. L. S., Riccobono, F., Richards, N. A. D., Rissanen, M. P., Rondo, L., Sarnela, N., Schobesberger, S., Sengupta, K., Simon, M., Sipilä, M., Smith, J. N., Stozkhov, Y., Tomé, A., Tröstl, J., Wagner, P. E., Wimmer, D., Winkler, P. M., Worsnop, D. R., and Carslaw, K. S.: Global atmospheric particle formation from CERN CLOUD measurements, Science, 354, 1119–1124, https://doi.org/10.1126/science.aaf2649, 2016.

Gordon, H., Kirkby, J., Baltensperger, U., Bianchi, F., Breitenlechner, M., Curtius, J., Dias, A., Dommen, J., Donahue, N. M., Dunne, E. M., Duplissy, J., Ehrhart, S., Flagan, R. C., Frege, C., Fuchs, C., Hansel, A., Hoyle, C. R., Kulmala, M., Kürten, A., Lehtipalo, K., Makhmutov, V., Molteni, U., Rissanen, M. P., Stozkhov, Y., Tröstl, J., Tsagkogeorgas, G., Wagner, R., Williamson, C., Wimmer, D., Winkler, P. M., Yan, C., and Carslaw, K. S.: Causes and importance of new particle formation in the present-day and preindustrial atmospheres, Journal of Geophysical Research: Atmospheres, 122, 8739–8760, https://doi.org/10.1002/2017JD026844, 2017.

Svenhag, C., Sporre, M. K., Olenius, T., Yazgi, D., Blichner, S. M., Nieradzik, L. P., and Roldin, P.: Implementing detailed nucleation predictions in the Earth system model EC-Earth3.3.4: sulfuric acid–ammonia nucleation, Geosci. Model Dev., 17, 4923–4942, https://doi.org/10.5194/gmd-17-4923-2024, 2024.

Weber, J., Archer-Nicholls, S., Abraham, N. L., Shin, Y. M., Bannan, T. J., Percival, C. J., Bacak, A., Artaxo, P., Jenkin, M., Khan, M. A. H., Shallcross, D. E., Schwantes, R. H., Williams, J., and Archibald, A. T.: Improvements to the representation of BVOC chemistry–climate interactions in UKCA (v11.5) with the CRI-Strat 2 mechanism: incorporation and evaluation, Geosci. Model Dev., 14, 5239–5268, https://doi.org/10.5194/gmd-14-5239-2021, 2021.

Weber, J., Archer-Nicholls, S., Abraham, N. L., Shin, Y. M., Griffiths, P., Grosvenor, D. P., Scott, C. E., and Archibald, A. T.: Chemistry-driven changes strongly influence climate forcing from vegetation emissions, Nat. Commun., 13, 7202, https://doi.org/10.1038/s41467-022-34944-9, 2022.